# Characterization of the pig lower respiratory tract antibiotic resistome

Yunyan Zhou[1,2,3], Jingquan Li[1,3], Fei Huang[1,3], Huashui Ai ®[1], Jun Gao[1], Congying Chen ®[1] ✉ & Lusheng Huang ®[1] ✉

Respiratory diseases and its treatments are highly concerned in both the pig industry and human health. However, the composition and distribution of antibiotic resistance genes (ARGs) in swine lower respiratory tract microbiome remain unknown. The relationships of ARGs with mobile genetic elements (MGEs) and lung health are unclear. Here, we characterize antibiotic resistomes of the swine lower respiratory tract microbiome containing 1228 open reading frames belonging to 372 ARGs using 745 metagenomes from 675 experimental pigs. Twelve ARGs conferring resistance to tetracycline are related to an MGE *Tn916* family, and multiple types of ARGs are related to a transposase gene *tnpA*. Most of the linkage complexes between ARGs and MGEs (the *Tn916* family and *tnpA*) are also observed in pig gut microbiomes and human lung microbiomes, suggesting the high risk of these MGEs mediating ARG transfer to both human and pig health. Gammaproteobacteria are the major ARG carriers, within which *Escherichia coli* harbored >50 ARGs and >10 MGEs. Although the microbial compositions structure the compositions of ARGs, we identify 73 ARGs whose relative abundances are significantly associated with the severity of lung lesions. Our results provide the first overview of ARG profiles in the swine lower respiratory tract microbiome.

Antibiotic resistance genes (ARGs) have been considered as emerging threats to global public health. Many studies have shown that antibiotics exposure leads to a decrease in microbial diversity[1,2], and many pathogens that could have been effectively eliminated by antibiotics are now no longer sensitive to these antibiotics, resulting in many multidrug-resistant (MDR) bacteria. Most of the studies concerning antibiotic resistance in humans and animals have focused on the gut microbiota. With the increasing prevalence of respiratory diseases, especially the global pandemic of Coronavirus disease 2019 (COVID-19) in recent years, increasing amounts of antibiotics have been used to treat respiratory diseases. Whether this will cause the accumulation of ARGs in the respiratory tract microbiota is unknown. Although there has been increased attention paid to the relationship between the lung microbiome and respiratory diseases[3], the association between the

lung antibiotic resistome and the composition of lung microbiota or the role of the lung resistome in the progression of diseases has rarely been reported.

Shuai et al. (2022) found that the composition of the human gut antibiotic resistome was associated with the progression of diabetes[4]. Another study suggested that ARGs could serve as potential predictors of autism spectrum disorder[5]. Whether ARGs are also related to respiratory diseases or can be used as the biomarkers for disease diagnosis is largely unknown. Several studies have revealed that the airways of humans with chronic respiratory diseases are important ARG reservoirs[6,7]. Owing to the difficulty in collecting human lung microbiota samples, most of the studies on respiratory diseases have only used nasopharyngeal swabs, sputum, blood, or fecal samples. However, these surrogate measurements are often insufficient to

[1]National Key Laboratory of Swine Genetic Improvement and Germplasm Innovation, Jiangxi Agricultural University, Nanchang 330045, China. [2]Institute of Engineering Biology and Health, Collaborative Innovation Center of Yangtze River Delta Region Green Pharmaceuticals, College of Pharmaceutical Sciences, Zhejiang University of Technology, Hangzhou 310014, China. [3]These authors contributed equally: Yunyan Zhou, Jingquan Li, Fei Huang. ✉e-mail: chcy75@hotmail.com; Lushenghuang@hotmail.com

reflect the actual physiological states of lungs or the biological nature of lung infections. Considering the similar compositions of some bacterial species in the lung microbiome between humans and pigs[8], the ARG profiles of pig lower respiratory tract microbiome should provide a reference for human lower respiratory tract microbiome.

Pigs as important domestic animals produce one-third of the meat consumed globally. Swine respiratory diseases are one of the main challenges to the pig industry because it can cause significant economic losses. Several studies have identified significant differences in the composition of lung microbial communities in pigs with different degrees of lung lesions that were associated with growth performance, meat quality, and other physiological and biochemical indicators[9,10]. The extensive use of antibiotics in the pig industry promotes the emergence and spread of ARGs. The long-term utilization of antibiotics for swine respiratory diseases could alter the composition of the lung microbiome and resistome. Therefore, understanding the composition of the antibiotic resistome in the pig lung microbiome could guide the use of antibiotics in treating swine respiratory diseases. However, to our knowledge, there are no systematic studies on the diversity and composition of ARGs in the swine lung microbiome, nor are there published reports on whether ARGs are related to lung lesions.

ARGs can be horizontally transferred within or between microbial communities through mobile genetic elements (MGEs) including insertion sequences, transposons, gene cassettes/integrons, plasmids, and integrative conjugative elements[11]. Currently, most of the studies on the relationships between mobilome and resistome have focused on the gut and environment microbiome. However, the diversity and composition of MGEs in the swine lung microbiome and their roles in shaping the lung resistome are unclear. Furthermore, antibiotic resistance leads to the increased abundance of ARB and pathogens. Some commensal bacteria have been transformed to pathogens under the selective pressure of antibiotics[12]. Thus, understanding the host bacteria of ARGs is critical for treating diseases and controlling the transmission of antibiotic resistance in pig farms and environments. *Mycoplasma hyopneumoniae* has been regarded as one of the primary pathogens associated with chronic respiratory illnesses in swine[13]. Controlling the infection of *Mycoplasma hyopneumoniae* is a major goal in swine production[14]. However, the pathogenic mechanism of *Mycoplasma hyopneumoniae* is not well understood. Understanding whether *Mycoplasma hyopneumoniae* harbors ARGs would be helpful for the treatment of *Mycoplasma hyopneumoniae* infection.

In this study, we used 745 lower respiratory tract metagenomes from 675 experimental pigs across five wild and domesticated pig populations to characterize the composition and distribution of antibiotic resistome (Supplementary Data 1). We investigated the relationship between ARGs and MGEs, and identified the host bacteria of ARGs in the swine lower respiratory tract microbiome. The antibiotic resistome profiles between bronchoalveolar lavage (BAL) fluid and tracheal lavage fluid samples were compared, and the relationship between ARGs and lung lesions was also evaluated. We found that *Mycoplasma hyopneumoniae* that had the highest abundance in the lung microbiome of experimental pigs did not carry ARGs but rather harbored virulence factors genes (VFGs) that showed an association with MGEs.

## Results

### Composition and distribution of ARGs in the pig lower respiratory tract microbiome

Using 745 metagenome data of the lower respiratory tract microbiome from 675 experimental pigs across five populations, we constructed a swine respiratory microbial gene catalog containing 10,337,194 non-redundant genes. Of these, 1228 open reading frames (ORFs) were identified as antibiotic resistance protein-coding genes by aligning against the Comprehensive Antibiotic Resistance Database (CARD).

These ORFs belonged to 372 ARGs (e.g., *tetQ*, *tet(39)*, *APH(6)-Id*, *ANT(6)-Ia*) and 24 antibiotic resistance types (e.g., tetracycline resistance, aminoglycoside resistance) according to the antibiotic classes to which they conferred resistance (Supplementary Data 2). To investigate the possible contamination introduced during sample collection, DNA extraction and sequencing, we sequenced twelve control samples and analyzed the composition of ARGs. Only three ARGs were identified in twelve control samples, including *adeF*, *TEM-181*, and *TEM-237*. Among them, only *adeF* was also identified in 745 experimental samples. The *adeF* identified in control samples contained only one ORF. In 745 experimental samples, a total of 57 ORFs were identified as *adeF*. We blasted the *adeF* ORF identified in control samples with 57 *adeF* ORFs in experimental samples. The highest sequence identity was only 72.3%, indicating different *adeF* ORFs between control samples and experimental samples. These results suggested that the contaminations are unlikely to have an influence on the ARGs profiles of experimental samples.

Among 372 ARGs detected in this study, 205 (55%) conferred resistance to only one drug class, and 167 (45%) conferred resistance to at least two antibiotic classes, among which 4% ARGs showed resistances to macrolide, lincosamide, and streptogramin antibiotics (MLS) (Fig. 1a). Tetracycline resistance (33%) was most enriched in the swine lower respiratory tract microbiome, followed by aminoglycoside (22%) (Fig. 1b). These two types of ARGs were also most abundant in the pig gut microbiome[15]. In addition, phenicol (14%) and multidrug (13%) resistance also had high abundances in the swine lower respiratory tract microbiome (Fig. 1b). Twelve ARGs were found in more than 75% of the samples. The prevalence values of *adeF*, *floR*, *tet(W/N/W)*, *APH(6)-Id*, and *tetQ* were the top five. However, more than half (57%) of the ARGs were detected in less than 10% of the samples. The ARGs whose abundance was ranked in the top five included *floR*, *tet(39)*, *tet(L)*, *tetQ*, and *tet(D)* (Fig. 1c and Supplementary Data 3). Antibiotic efflux (39.8%) and antibiotic inactivation (33.3%) were the major resistance mechanisms of ARGs identified in the swine lower respiratory tract microbiome (Supplementary Fig. 1a).

We then analyzed the distribution of 10 clinically important ARGs in pig lower respiratory tract microbiome. These 10 ARGs included mobile colistin resistance (*mcr*) gene, two tetracycline resistance genes (*tet(L)* and *tet(D)*), one sulfonamide resistance genes (*sul1*), two MLS resistance genes (*ErmB* and *ErmT*), and four aminoglycoside resistance genes (*APH(6)-Id*, *APH(3″)-Ib*, *ANT(6)-Ia*, and *aad(6)*). Among them, *sul1* and *ErmB* were highly variable in pig lower respiratory tract microbiome. Six and seven ORFs were identified for *sul1* and *ErmB*, respectively. The prevalence of these ORFs varied in 745 tested samples (Supplementary Table 1). We found 13 subtypes of *mcr* that was first identified in pigs, including *mcr-1*, *mcr-1.2*, *mcr-3.4*, and *mcr-4*. The prevalence of these *mcr* subtypes was different in 745 tested microbial samples (Supplementary Table 1).

### Mobile genetic elements related to antibiotic resistance genes

MGEs play a central role in the horizontal transfer of antibiotic resistance genes between bacterial cells. It is important to understand the distribution of MGEs in bacteria and their relationship with ARGs. A total of 3016 ORFs belonging to MGEs were identified from 745 metagenomes by aligning protein sequences of gene catalog against the MGE Database "MobileGeneticElementDatabase" created by Parnanen, et al.[16] (Supplementary Data 4). These 3016 ORFs belonged to 83 MGEs and were classified into 23 MGE types including transposase, *IS91*, *Tn916*, *istB*, *istA*, integrase, plasmid, and qacEdelta (Supplementary Fig. 1b). The richness of MGEs in tested samples was significantly related to the α-diversity of ARG composition (Supplementary Fig. 1c, d). Transposase genes containing 27 MGEs had the highest abundance in swine lower respiratory tract microbiome, accounting for 79% of the total abundance of MGEs (Supplementary Fig. 1b and Supplementary Data 4). A total of 105 ORFs covering 19 MGE types were identified as

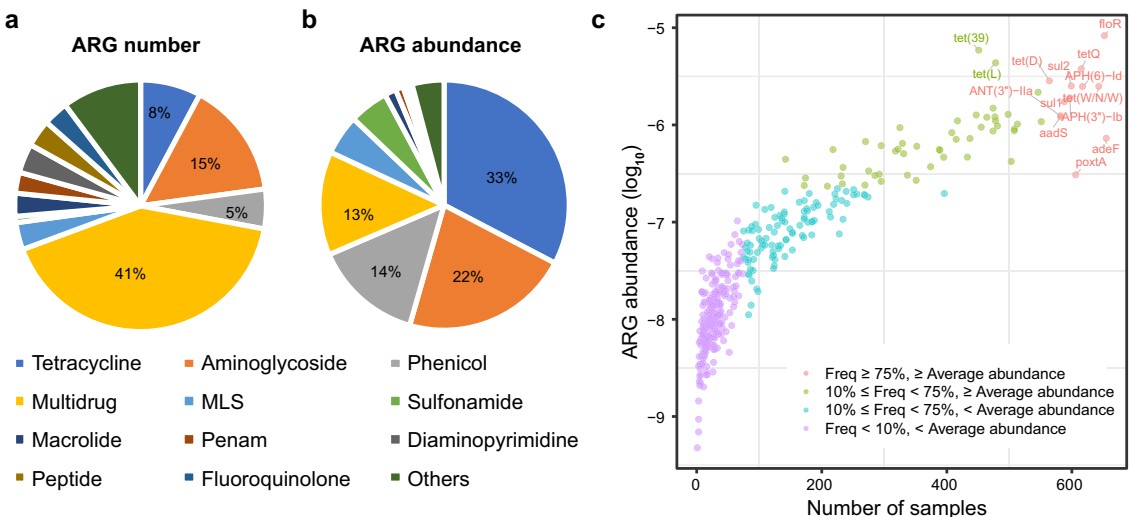

**Fig. 1 | Composition, abundance, and distribution of antibiotic resistance genes (ARGs). a** The proportion of the number of unique ARGs in each ARG type to the total number of ARGs. **b** The proportion of the abundance of ARGs in each ARG type to the total abundance of ARGs. **c** Distribution of prevalence and abundances of all ARGs in tested samples (n = 745). The abundance of each ARG was defined as the average abundance of that gene in all tested samples. Different color dots indicate the distribution of ARGs according to their abundances and frequencies. Source data are provided as a Source Data file.

plasmid genes. However, these plasmid genes only accounted for 1.0% of the total abundance of all MGEs (Supplementary Fig. 1b and Supplementary Data 4).

At the contig level, only 16 ARG ORFs co-occurred with MGEs on the same contigs. This low co-occurrence rate should be due to the short contig length, since the average length of bacterial genes is about 1000 bp[17], and the average length of contigs in the current study was only 1136 bp due to the low sequencing depth. This meant that the relationships between ARGs and MGEs only based on the co-occurrence on the same contigs were massively underestimated. Therefore, we explored the co-abundance relationships between ARGs and MGEs, and found significant correlations between 59 ARGs and 25 MGEs in their abundances ($r \geq 0.5$, FDR < 0.05, Spearman correlation) (Fig. 2). Twelve ARGs showing resistance to tetracycline, such as *tet(W/N/W)*, *tetQ*, and *tetM* were significantly associated with the *Tn916* family (*Tn916-orf13*, *Tn916-orf14*, *Tn916-orf15*, *Tn916-orf16*, *Tn916-orf18*, and *Tn916-orf7*) that is one of the types of conjugative elements transferring ARGs between bacterial cells[18]. Previous report suggested that the close proximity of MGEs and ARGs ( < 5.0 kb) is more likely to induce horizontal gene transfers (HGTs)[19]. Interestingly, the close linkages (or linkage complexes) between the *Tn916* family and *tetM* were located within <5 kb of region on the same contigs and observed in two metagenome assembled genomes (MAGs). These two MAGs were classified as Moraxellaceae and *Jeotgalicoccus schoeneichii*, and detected in 33 and 161 out of 745 tested samples, respectively (Fig. 3a). The *floR* gene that had the highest abundance in the swine lung microbiome (Fig. 1c) was also associated with multiple MGEs belonging to the *Tn916* family. The ARGs conferring resistance to aminoglycosides, including *aadA27*, *APH(6)-Id*, *APH(3")-Ib*, *ANT(3")-IIc* and *ANT(3")-IIa*, were significantly associated with multiple MGEs. The *adeF* was significantly associated with 13 MGEs, including five transposase genes (*tnpA*, *tnpA1*, *tnpA3*, *tnpA5*, and *tnpAcp2*) (Fig. 2). This means that these ARGs may be transferred horizontally by different types of MGEs, explaining why these ARGs had high prevalence and abundance in tested samples. *TnpA*, a transposase gene, was associated with 20 resistance genes conferring resistance to multiple antibiotic types including seven aminoglycoside resistance genes (*ANT(3")-IIa*, *ANT(3")-IIc*, *APH(3')-Ia*, *APH(3")-Ib*, *APH(6)-Id*, *aadA27* and *aadA5*), four tetracycline resistance (*tet(39)*, *tet(D)*, *tet(W/N/W)* and *tetQ*), and two sulfonamide resistance genes (*sul1* and *sul2*) (Fig. 2). According to previous study, the ability of HGTs of ARGs across bacterial genomes can be

measured by the number of the related MGEs, and HGTs more likely occur in those ARGs that are closely linked with MGEs[20]. We found that *TnpA* was indeed close proximity to various types of ARGs, especially aminoglycoside resistance genes, based on the contigs of MAGs. For example, the *tnpA* was in close proximity to *sul2* on a contig of MAG_75 (*Acinetobacter towneri*) that were detected in 123 out of 745 tested samples (Fig. 3b). This suggested that *tnpA* should contribute to the HGT of multiple ARGs. *repUS12*, a MGE belonging to plasmid, was significantly associated with 6 ARGs including *ANT(4')-Ib*, *ANT(6)-Ia*, *APH(2")-If*, *ErmT*, *tet(45)* and *tet(L)* (Fig. 2). Among them, *tet(L)* coexisted with *repUS12* on the same contig (Supplementary Fig. 2).

To further confirm the abundance changes of ARGs, MGEs and MAGs in tested samples and the close relationships between ARGs and MGEs, and between ARGs, MGEs and MAGs carrying these genes, we performed the qPCR for ARGs *tetM*, *APH(6)-Id* and *ANT(3")-IIa*, MGEs *Tn916-orf13* and *Int-Tn916*, and MAGs MAG_21 (Moraxellaceae), MAG_26 (*Jeotgalicoccus_A schoeneichii*) and MAG_340 (*Escherichia coli*) in 23 samples that were detected and undetected the abundances of these ARGs, MGEs and MAGs. The results found the significant correlations between the abundance values from qPCR (ΔCt) and the abundances from metagenomic sequencing for near all ARGs and MGEs, except *tetM* which showed the tendency of correlations but not achieve significance level (R = 0.35, P = 0.099) (Supplementary Fig. 3a). For three MAGs, the abundance changes were well repeated between qPCR and metagenomic sequencing analysis in 23 samples for MAG_26 and MAG_340 although the MAG_21 did not fit well. Furthermore, as we expected, the close relationships between ARGs and MGEs, and between ARGs, MGEs and MAGs carrying these genes were well confirmed in the qPCR (Supplementary Fig. 3b).

## Potential horizontal transfers of ARGs cross pig lung and gut microbiome, and human lung microbiome

To investigate potential horizontal transfers of ARGs mediating by MGEs cross microbiomes in different body sites (lung and gut) and between pigs and humans, we analyzed the linkage relationships between ARGs and MGEs using 6339 MAGs recovered from 500 pig gut metagenomes from our previous study[21]. The *tnpA* and *Tn916* family were the dominant MGEs in the pig gut microbiome. The *Tn916* family (*Tn916-orf8* and *Tn916-orf9*) was closely linked to the *tetM* that confers resistance to tetracycline on the same contig (Fig. 3a). In addition, the *tnpA* was also detected to be in close proximity to various ARG types in

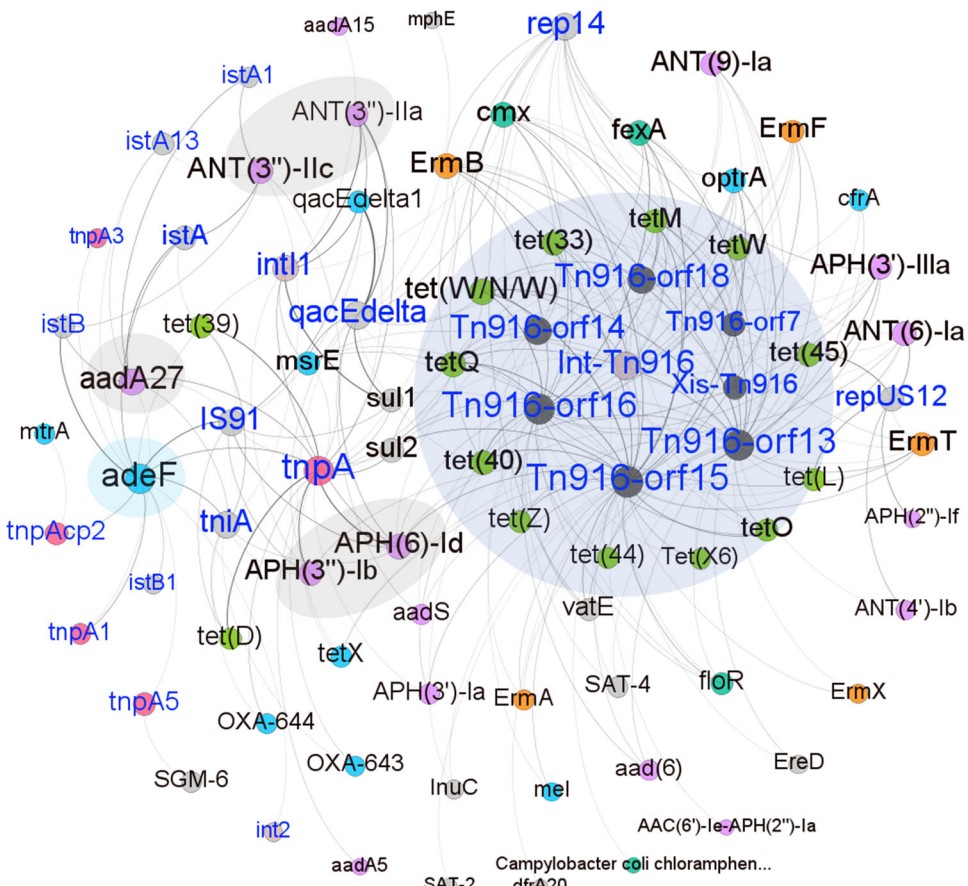

**Fig. 2 | Co-abundance network between antibiotic resistance genes (ARGs) and mobile genetic elements (MGEs).** The connections between ARGs and MGEs with *Spearman* correlation coefficients ≥ 0.5 and *P* values < 0.05 are shown in the figure. The names of MGEs are marked in blue and the names of ARGs are marked in black. The nodes with the larger text size mean more connections of that gene with other genes. Nodes are colored according to ARG or MGE types. *Tn916* family, twelve ARGs conferring resistance to tetracycline and related to *Tn916* family, and *adeF* and five ARGs conferring resistance to aminoglycosides, all of which were correlated with multiple MGEs are highlighted with different colored shadows. Source data are provided as a Source Data file.

the pig gut microbiome, especially, to aminoglycoside resistance genes with high frequency (Fig. 3b).

To evaluate potential horizontal transfers of ARGs between pigs and humans, which should bring high risk to human health, we further investigated the close linkages between ARGs and MGEs in the human lung microbiome. We used 46 metagenomic sequencing data of BAL fluid samples from children with pneumonia and 118 metagenomic sequencing data of BAL fluid samples from adult COVID-19 patients. Similar to that in pig lung and gut microbiomes, the *adeF* and *tnpA* were the most prevalent and dominant ARG and MGE, respectively, in the human lung microbiome. Moreover, in human lung microbiome, the *tnpA* was found to closely link to different ARG types including aminoglycoside resistance genes (*APH(3')-IIIa, ANT(4')-Ib* and *APH(6)-Id*), MLS resistance genes (*ErmB*) and phenicol resistance genes (*QnrS1*), which also present in the gut microbiome of pigs (Fig. 3b). The close linkages between the *Tn916* family and *tetM* was also verified in the human lung microbiome. For examples, *Int-Tn916, Xis-Tn916, Tn916-orf7, Tn916-orf8, Tn916-orf9* were tightly linked to *tetM* on the same contig which was classified as *Streptococcus* from the lung microbiome of children with pneumonia (Fig. 3a). *tetM* were also in close proximity to the Tn916 family including *Tn916-orf9, Xis-Tn916, Tn916-orf13, Tn916-orf14, Tn916-orf15, Tn916-orf16, Tn916-orf17, Tn916-orf18, Tn916-orf19, Tn916-orf120* in three contigs from the lung microbiome of adult COVID-19 patients.

The co-occurrence relationships between *tetM* and the *Tn916* family, and between *tnpA* and various types of ARGs were only detected in a few contigs in both human and pig lower respiratory

tract metagenome data used in this study. This should be due to the short contig lengths that were caused by the low sequencing depth. To confirm that these relationships extensively existed, but not an especial case, we downloaded 3878 sequenced genomes of isolated common antibiotics resistance bacterial species from humans and pigs in the Refseq Database, including 1172 *Escherichia coli*, 529 *Acinetobacter baumannii*, 642 *Pseudomonas aeruginosa*, 976 *Staphylococcus aureus*, 168 *Enterococcus faecalis*, 297 *Enterococcus faecium*, and 94 *Streptococcus suis* (Supplementary Data 5). We observed that *tetM* widely coexisted with the *Tn916* family in various bacterial species, including *Enterococcus faecalis* (existed in 105 out of 168 genomes, 105/168), *Enterococcus faecium* (179/297), *Streptococcus suis* (14/94), *Staphylococcus aureus* (124/976), and *Escherichia coli* (38/1,172) (Supplementary Table 2 and Supplementary Fig. 4a). Notably, not all strains of these bacterial species carried *tetM*, but near all genomes having *tetM* also carried the *Tn916* family. This result further confirmed the suggestion that the *Tn916* family might play an important role in the horizontal transfer of *tetM*. Besides *tetM*, another tetracycline ARG *tet(45)* was also related to the *Tn916* family and always adjacent to *tetM* in *Enterococcus faecalis* and *Enterococcus faecium*. And *tet(W/N/W)* also co-occurred with the *Tn916* family in several strains of *Enterococcus faecalis* and *Enterococcus faecium* (Supplementary Fig. 4a). A total of 78,514 MGE ORFs were identified in these 3878 genomes, 72.5% of which were *tnpA* that belongs to transposase. This was consistent with the result that transposase genes had the highest abundance (79.0%) in swine lower respiratory tract microbiome (Supplementary Fig. 1b). *tnpA* was in close proximity to various types of ARGs (Supplementary

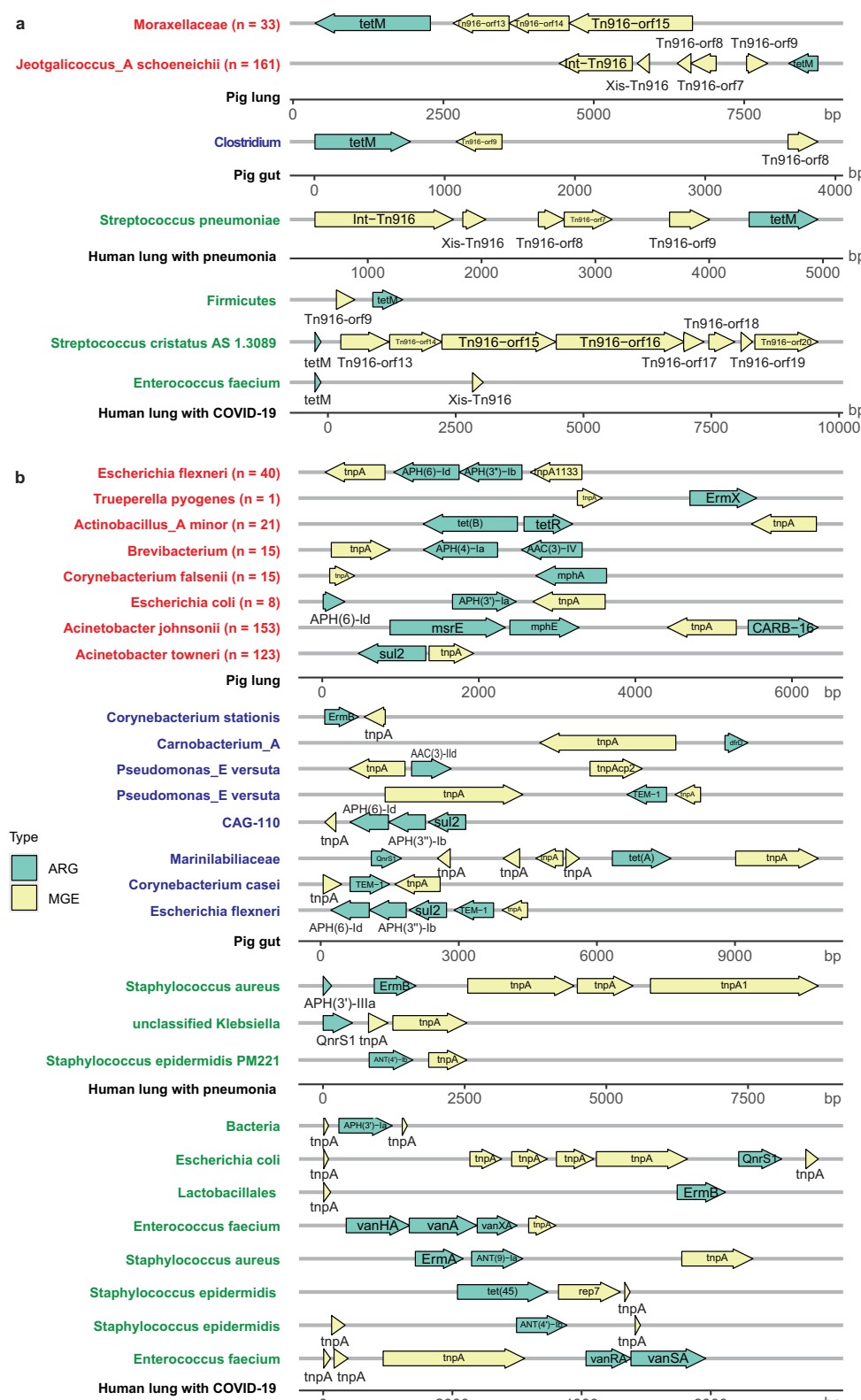

**Fig. 3 | Exhibition of tight linkages between antibiotic resistance genes (ARGs) and mobile genetic elements (MGEs) based on contigs. a** The close linkages between *tetM* and *Tn916* family in the MAG contigs from pig lung microbiome (labeled with red color in the *y*-axis), pig gut microbiome (labeled with blue color in the *y*-axis), and human lung microbiome (labeled with green color in the *y*-axis). **b** The close linkages between *tnpA* and various ARGs in the MAG contigs from pig lung microbiome (labeled with red color in the *y*-axis), pig gut microbiome (labeled with blue color in the *y*-axis), and human lung microbiome (labeled with green

color in the *y*-axis). The *x*-axis represents the location of genes in the contig. Green arrows indicate ARGs, and yellow arrows indicate MGEs. The directions of arrows represent the strand on which genes are located. Right arrows indicate the genes on the forward strand, and the left arrows indicate the genes on the reverse strand. The *y*-axis indicates the bacterial taxa annotated to MAGs containing the corresponding contigs. The numbers in brackets show the sample numbers detected the corresponding MAG. Source data are provided as a Source Data file.

Fig. 4b), providing the evidence that *tnpA* might contribute to the HGT of multiple ARGs.

Taken together, the close linkages between *Tn916* family and *tetM*, and between *tnpA* and various ARGs were detected in different bacterial species of all human and pig lung microbiomes, and pig gut microbiome. These results suggested that the *Tn916* family and *tnpA* might mediate the HGTs of ARGs among different body sites, and between humans and pigs through different bacterial species, and pigs might be used as a model for studying the MGE-mediated horizontal transfers of ARGs in humans from which it was difficult to obtain microbial samples of lower respiratory tract.

## Host bacteria of ARGs

We analyzed the bacterial compositions of swine lower respiratory tract microbiome in another our study[8]. In brief, a total of 81 phyla, 1018 genera, and 1611 species were identified in 745 tested samples. Proteobacteria (44%), Tenericutes (31%), Firmicutes (10%), Bacteroidetes (6%), and Actinobacteria (4%) were the predominant phyla of swine lower respiratory tract microbiome (Supplementary Fig. 5a). *Mycoplasma* (40%) and *Acinetobacter* (17%) were the two most abundant bacterial genera (Supplementary Fig. 5b). At the species level, *Mycoplasma hyopneumoniae* had the highest abundance in the swine lower respiratory tract microbiome, followed by *Acinetobacter johnsonii*, and *Escherichia coli*. Most of the ESKAPE pathogens including *Enterococcus faecium*, *Staphylococcus aureus*, *Klebsiella pneumoniae*, *Acinetobacter baumannii*, *Pseudomonas aeruginosa*, and *Enterobacter* spp., which have been regarded as the most troublesome pathogens in hospitals due to their high frequency of resistance to antibiotics, were also identified in the swine lower respiratory tract microbiome (Supplementary Fig. 5c).

ARGs endow their host bacteria with antibiotic resistance, and they may even lead to the production of MDR bacteria. Here, the host bacteria of ARGs were determined by the taxonomic assignment of the contigs harboring the ORFs of ARGs. A total of 1228 non-redundant ARG ORFs were distributed in 1209 contigs, of which 1183 were annotated to bacteria (Supplementary Data 2). The bacteria from Proteobacteria (mainly from Gammaproteobacteria) were the major carriers of ARGs, harboring 53% of ARGs (Fig. 4a). These bacteria also had the highest relative abundance in swine lower respiratory tract microbiome (Supplementary Fig. 5c). However, no ARGs were detected in the Tenericutes that accounted for 31% of the total abundance of lung microbiome (Fig. 4a and Supplementary Fig. 5c). *Pseudomonas aeruginosa* harbored the largest number of ARGs (61 ARG ORFs), followed by *Escherichia coli* (52 ARG ORFs). Fourteen species from *Acinetobacter* were found to harbor a total of 97 ARGs (Supplementary Data 2). Nearly half of the bacterial species that were detected in nearly all samples and whose abundances were listed in the top 20 in the swine lower respiratory tract microbiome, belonged to Proteobacteria. These species from Proteobacteria harbored a large number (4–52) of ARGs (Supplementary Fig. 5c).

We further focused on the distribution of host bacteria for the five ARGs with the highest abundances, including *floR*, *tet(39)*, *tet(L)*, *tet(Q)* and *tet(D)*. As expected, all of these ARGs had a wide range of host bacteria even across different phyla (Fig. 4b). For example, the *floR* gene conferring resistance to phenicol antibiotics was detected in three bacterial species, *Escherichia coli* (Proteobacteria), *Providencia rettgeri* (Proteobacteria), and *Chryseobacterium* sp. *POL2* (Bacteroidetes) (Supplementary Fig. 5d). This suggested that these ARGs might be involved in HGT across different bacterial species across phyla.

To characterize the host bacteria of ARGs at the strain level, we further grouped assembled contigs into metagenome-assembly genomes (MAGs), resulting in 397 non-redundant MAGs from 745 metagenomes (GWHBPMO00000000 ~ GWHBQBU00000000, https://ngdc.cncb.ac.cn/bioproject/browse/PRJCA010893). A total of 416

ARG ORFs corresponding to 152 ARGs were detected in 115 MAGs (Supplementary Data 6). Sixty-two MAGs carried more than two ARGs. Among these, 11 MAGs carried more than five ARGs (Supplementary Fig. 6a). Three *Escherichia coli* MAGs (MAG_340, MAG_368, and MAG_389) harbored more than 50 ARGs (Supplementary Fig. 6b). The average of relative abundances of MAG_340, MAG_368, and MAG_389 in the samples detected these MAGs were 0.15% (0.01%-1.37%), 0.43% (0.01%-2.79%), and 0.38% (0.002%-2.23%), showing a relatively high abundance although their prevalence was low (only detected in 40, eight and eight out of 745 experimental samples). In addition, multiple homologous gene clusters encoding ARGs were identified in these three *Escherichia coli* MAGs, including the clusters *emr*, *mdt*, and *Acr* families (Supplementary Data 7). These results further explained the super antibiotic resistance of *Escherichia coli*. There were 88 ARGs that existed in more than two host bacteria (Supplementary Fig. 6c and Supplementary Data 6). There were nine ARGs, including *adeF*, *tet(39)*, and *floR*, whose abundance or prevalence was ranked in the top five in all detected ARGs (Fig. 1c) and that were identified in more than five host bacteria (Supplementary Fig. 6d). The *tet(39)* and *aadA27* genes were mainly carried by *Acinetobacter johnsonii*, whereas *adeF*, *rsmA*, and *floR* were harbored by various bacterial species (Supplementary Fig. 6d).

To evaluate the risk of horizontal transfer of ARGs at the strain level, we further analyzed the distribution of MGEs in three *Escherichia coli* MAGs (MAG_340, MAG_368, and MAG_389) that carried more than 50 ARGs. These three MAGs harbored 35, 17 and 10 MGEs, respectively. The *tnpA* element, one of the transposase genes, found in multiple locations in each of the three MAGs, co-occurred with different ARGs on the same contigs. For example, a gene cluster was found within 22-kb of a contig in the MAG_340, where nine ARGs and 12 MGEs including 10 *tnpA* were distributed (Fig. 4c). Considering the relatively high abundance of MAG_340, the result suggested the high risk of horizontal transfer of these ARGs under the assistance of MGEs.

## Comparison of antibiotic resistome and MGE profiles between lung and trachea and among five pig populations

To compare the compositions of antibiotic resistomes between lung and trachea microbiomes, we used 138 metagenomic sequencing data of BAL fluid and tracheal lavage fluid samples from 69 pigs. The results showed that the number of ARGs (richness), the abundance of total ARGs, and the α-diversity (Shannon index) of resistome profile in the trachea microbiome was significantly higher than that in the lung microbiome (Fig. 5a–c), and 78% of ARGs ($n = 292$) had a higher prevalence in tracheal lavage fluid samples compared to BAL fluid samples (Supplementary Data 8). Principal co-ordinates analysis (PCoA) also found the significant difference in the compositions of ARGs between lung and trachea microbiomes (ANOSIM, $R = 0.045$, $P = 2.0 \times 10^{-3}$) (Supplementary Fig. 7a). We further identified 40 ARGs showing significantly different abundances between lung and trachea microbiome. Most of these differential ARGs had significantly higher abundance and prevalence in tracheal lavage fluid samples, and mainly conferred the resistance to multiple drugs or aminoglycosides (Fig. 5d). Nine OXA ARGs belonging to beta-lactamases showing resistance to multiple drugs and that were transferred by plasmids[22] were found in most of the tracheal lavage fluid samples but were almost absent in BAL fluid samples (Fig. 5d). The similar results were also identified in the MGE profiles. Significantly different MGE compositions were found between lung and trachea microbiomes (Supplementary Fig. 7b). Trachea microbiome had the higher number of MGEs (richness), total abundance of MGEs, and the α-diversity (Fig. 5e–g).

We further compared ARG and MGE profiles in pig lower respiratory tract microbiome among five pig populations. Overall, the richness, the total abundance, and the α-diversity (Shannon index) of ARGs were not significantly different among five pig populations

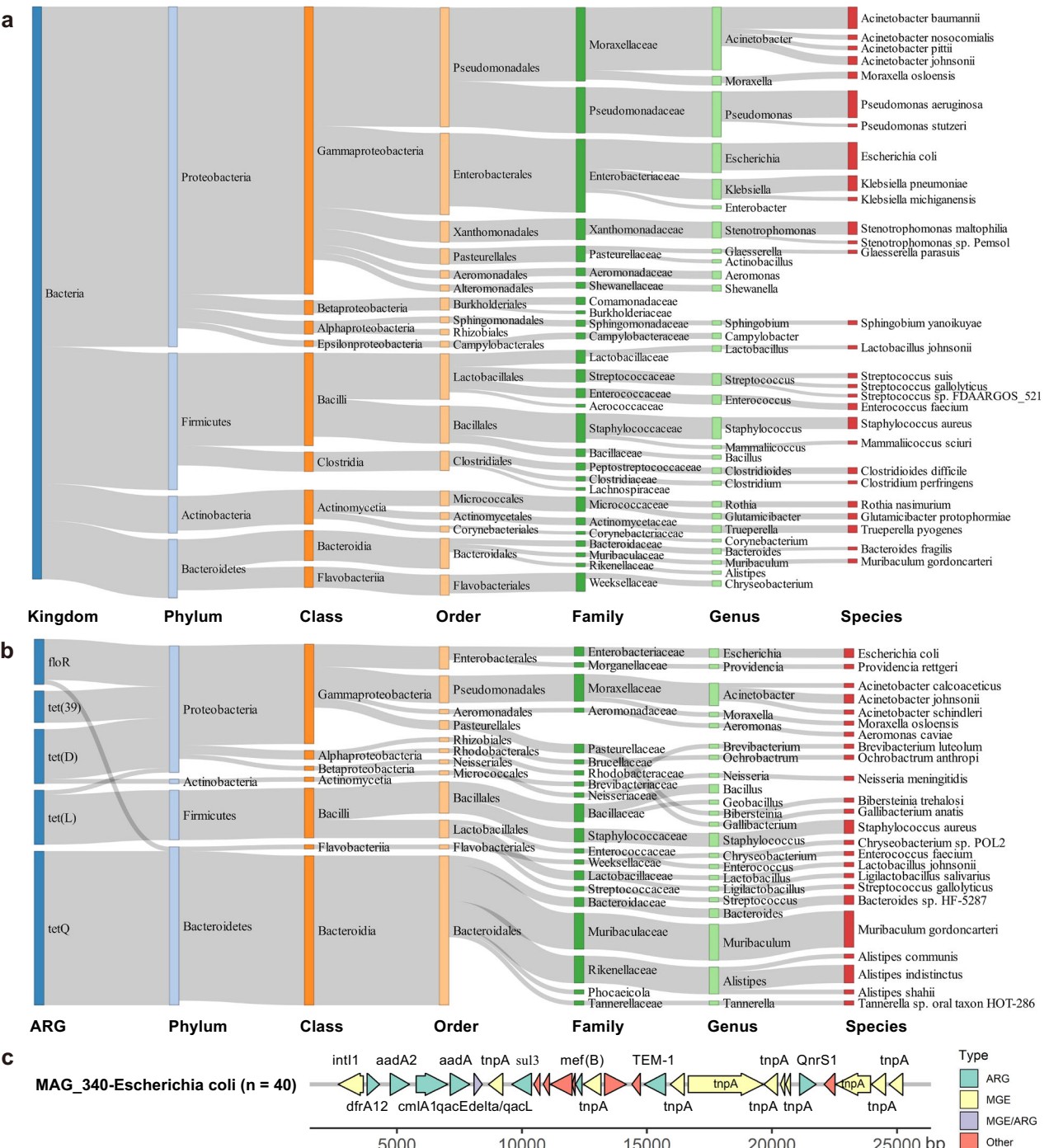

**Fig. 4 | Host bacteria of antibiotic resistance genes (ARGs) in swine lower respiratory tract microbiome. a** Distribution of host bacteria of 1209 contigs carrying 1228 ARG open reading frames (ORFs) at different taxonomic levels. The colors of the rectangles represent different taxonomy levels. The lengths of the rectangles indicate the number of contigs carrying ARGs. Only those taxa with the number of contigs carrying ARGs greater than five are shown. **b** Distribution of host bacteria of five ARGs with the relative abundance listed in the top five at different taxonomic levels. **c** The distribution of ARGs and MGEs in *Escherichia coli* MAG (MAG_340). The *x*-axis represents the location of genes in the contig. The directions of arrows represent the strand on which genes are located. Right arrows indicate the genes on the forward strand, and the left arrows indicate the genes on the reverse strand. Source data are provided as a Source Data file.

(Supplementary Fig. 8a–c). However, different pig populations showed distinct β-diversity in the ARG composition. Erhualian pigs had the highest β-diversity (Supplementary Fig. 9a). PCoA also indicated the distinct compositions of ARGs among five pig populations (Supplementary Fig. 9b). Based on the relative abundances, the $F_7$ population had the highest abundance of *floR*. *tet(L)* was the dominant ARGs in both Berkshire × Licha line and wild boars. *tet(39)* and *tetQ* were the dominant ARGs in Erhualian and Tibetan pigs, respectively.

The similar results were obtained in the compositions of MGEs among five pig populations. There were no significant differences in the richness, the total abundance, and the α-diversity of MGE compositions among five pig populations although distinct compositions of MGEs were observed through PCoA and β-diversity analysis (Supplementary Fig. 8d–f and Supplementary Fig. 9c, d). Indistinctively, *tnpA* had the highest abundance in all five populations, followed by *IS91*.

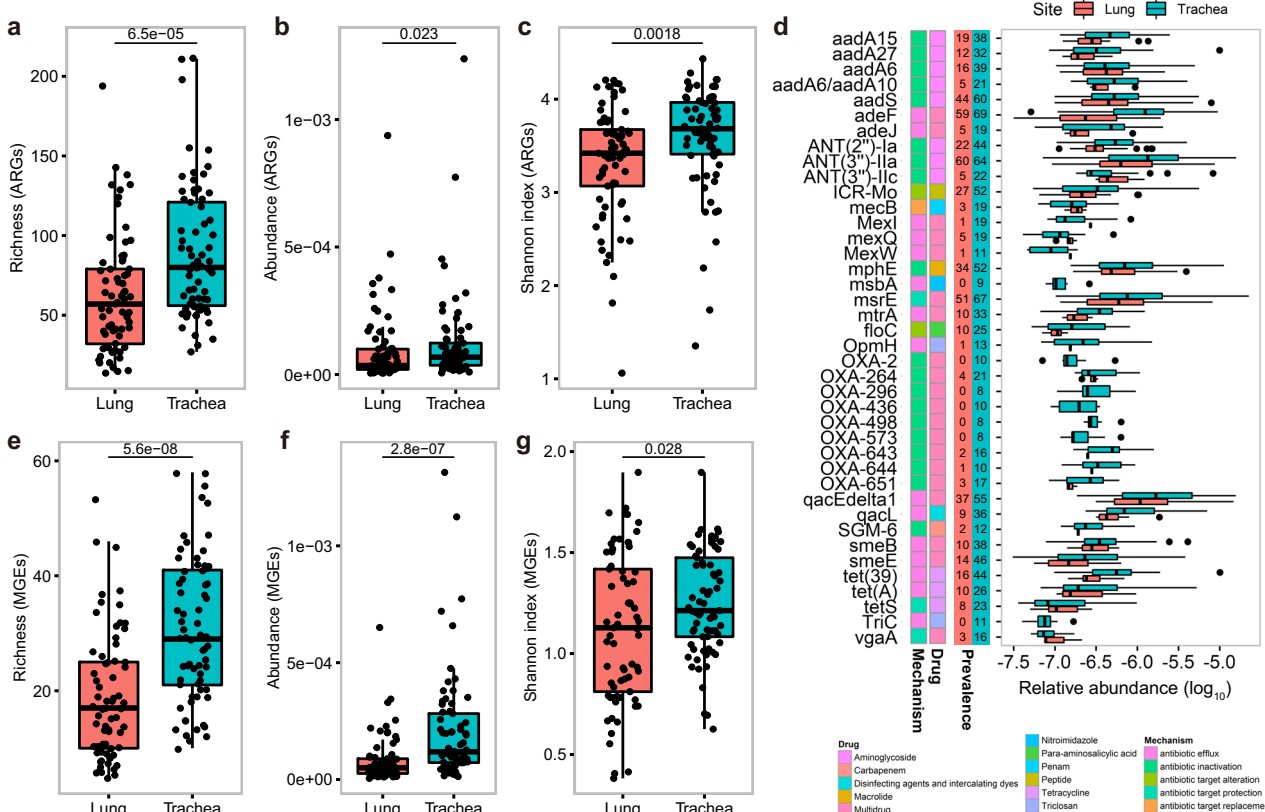

**Fig. 5 | Comparison of the antibiotic resistome between lung and trachea microbiomes. a** Comparison of the richness (number) of ARGs between lung ($n = 69$) and trachea microbiomes ($n = 69$). **b** Comparison of the abundance of total ARGs between lung ($n = 69$) and trachea microbiomes ($n = 69$). **c** Comparison of the evenness (Shannon index) of ARGs between lung ($n = 69$) and trachea microbiomes ($n = 69$). **d** Forty ARGs with significantly differential abundances between lung ($n = 69$) and trachea microbiomes ($n = 69$). The resistance mechanisms and resistance types of these ARGs are shown with different colors on the $y$-axis. The prevalence of ARGs in lung (red) and trachea (green) samples is also shown on the $y$-axis. The $x$-axis indicates the $\log_{10}$-transformed relative abundances. **e** Comparison of the richness (number) of MGEs between lung ($n = 69$) and trachea microbiomes ($n = 69$). **f** Comparison of the abundance of total MGEs between lung ($n = 69$) and trachea microbiomes ($n = 69$). **g** Comparison of the evenness (Shannon index) of ARGs between lung ($n = 69$) and trachea microbiomes ($n = 69$). All the comparisons were performed with bronchoalveolar and tracheal lavage fluid samples from the same 69 pig individuals using the two-sided Wilcoxon test. $P < 0.05$ corrected for multiple tests (FDR) was set as the significance threshold. Boxplots show median, 25th, and 75th percentile. The lower and upper boundaries of whiskers indicate the minima and maxima, respectively. The points laying outside the whiskers of boxplots represent the outliers. Source data are provided as a Source Data file.

## Relationship between ARGs in the lung microbiome and lung lesions

We systematically analyzed the relationships of lung microbial taxa and VFGs with pig lung lesions, and identified several lung bacterial taxa and VFGs that were significantly associated with the lung lesion levels[8]. Here, to evaluate the relationship of the diversity and composition of ARGs in the lung microbiome with lung lesions, we compared the profiles of lung resistome under different lung lesion levels using 613 BAL fluid samples from $F_7$ pigs of the mosaic population. Unexpectedly, the $\alpha$-diversity of ARG composition was decreased with the increased severity of lung lesions (Supplementary Fig. 10a, b). Considering the decreased $\alpha$-diversity of the lung microbial composition with the increased severity of lung lesions (Supplementary Fig. 10c, d), we examined whether the changes in the $\alpha$-diversity of ARGs (richness and the Shannon index in the ARG composition) were related to the shifts in the microbial composition. The result showed that the $\alpha$-diversity of ARG composition was positively associated with the $\alpha$-diversity of the microbial composition. Moreover, compared to the evenness (Shannon index) ($r = 0.27-0.3$, $P = 2.3 \times 10^{-14}-8 \times 10^{-12}$, the richness of bacterial species had a greater effect on the $\alpha$-diversity (richness and the Shannon index) of ARG composition ($r = 0.69-0.86$, $P < 2.2 \times 10^{-16}$) (Fig. 6a, b and Supplementary Fig. 10e, f). Similar results were also observed between the $\alpha$-diversity of the lung antibiotic resistome composition

and microbial gene richness (MGR) ($r = 0.38-0.43$, $P < 2.2 \times 10^{-16}$) (Supplementary Fig. 10g, h), confirming the effect of microbial composition on the $\alpha$-diversity of the ARG composition. The results of a Procrustes analysis indicated that the $\beta$-diversity of ARG composition was weakly correlated with the $\beta$-diversity of microbial composition ($r = 0.15$, $P = 1.0 \times 10^{-3}$) (Supplementary Fig. 11a). However, when *Mycoplasma hyopneumoniae* that had the highest abundance in the lower respiratory tract microbiome but did not carry ARGs was removed from the analysis, the samples with similar microbiome profiles tended to have more similar ARG profiles ($r = 0.57$, $P = 1.0 \times 10^{-3}$) (Supplementary Fig. 11b). These results suggested that the composition of lower respiratory tract microbial community, but not the degree of lung lesion severity structured the composition of swine lower respiratory tract resistomes. However, based on their relative abundances of ARGs, we indeed found that the abundance of ARGs conferring resistance to phenicol was significantly increased in pigs with severe lung lesions, while the abundance of ARGs for aminoglycosides was decreased with the severity of lung lesions (Fig. 6c). We further identified 73 ARGs showing significantly different abundances among pigs with different severities of lung lesions (*FDR* < 0.05). Only *floR* and *Haemophilus influenzae* PBP3 conferring resistance to beta-lactam antibiotics (*Hinf_PBP3_BLA*) were significantly enriched in pigs with severe lung lesions. There were 31, 32, and 8 ARGs that were enriched in pigs with healthy

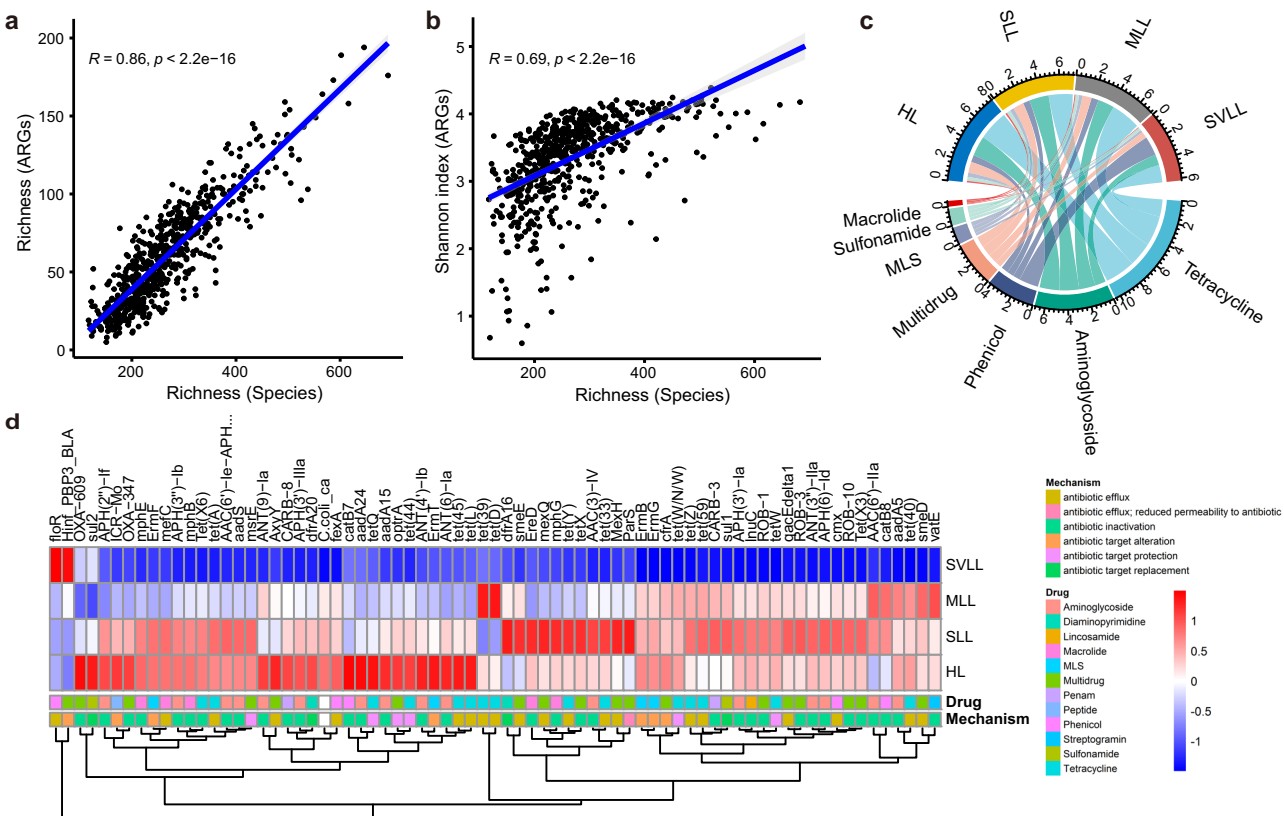

**Fig. 6 | Relationship of ARGs with lung lesions. a** Correlation between the richness (number) of bacterial species in the lower respiratory tract microbiome and the richness (number) of ARGs. **b** Correlation between the richness (number) of bacterial species in the lower respiratory tract microbiome and the evenness (Shannon index) of ARGs. The correlation analyses were performed in bronchoalveolar lavage fluid samples from $F_7$ pigs of a mosaic population ($n = 613$). The two-sided *Spearman* rank correlation tests and FDR corrections were performed using the *psych* R package. Data are presented as the actual value of the corresponding variable obtained for each sample (each point). **c** Composition and abundance of each ARG type in the lower respiratory microbiome with different lung lesion levels. HL, healthy lung (blue, $n = 51$); SLL, slight lung-lesion (yellow,

$n = 217$); MLL, moderate lung-lesion (gray, $n = 218$); SVLL, severe lung-lesion (dark red, $n = 127$). The abundances of ARG types shown in the Circos plot (the numbers in circles) are equal to the relative abundance × $10^5$. Different colored lines represent ARG types (antibiotics that ARGs show resistance to). **d** Seventy-three ARGs with significantly differential abundances among four pig groups with different lung lesions. The pairwise comparisons were performed using the two-sided Wilcoxon test, and a $P < 0.05$ corrected for false discovery rate (FDR) was treated as the significance threshold. The ARGs with prevalence <10% and average abundance <0.1% in tested samples were filtered. Resistance mechanisms and resistance types (drug) of these ARGs are shown with different color boxes on the horizon. Source data are provided as a Source Data file.

lungs, slight lung lesions, and moderate lung lesions, respectively, based on the average abundance of ARGs in each group (Fig. 6d).

**Putative horizontal transfer of VFGs mediated by MGEs in *Mycoplasma hyopneumoniae*, a pathogen causing lung diseases**
*Mycoplasma hyopneumoniae* is the main pathogen causing swine chronic respiratory diseases. Our study also found a significant increase in the abundance of *Mycoplasma hyopneumoniae* with the severity of lung lesions (Supplementary Fig. 11c). However, *Mycoplasma hyopneumoniae* has been reported to be sensitive to various types of antibiotics such as tetracyclines, macrolides, and aminoglycosides[23]. More importantly, no ARGs were detected in *Mycoplasma hyopneumoniae* genomes, including seven MAGs recovered in this study and 23 genomes downloaded from NCBI RefSeq databases although *Mycoplasma hyopneumoniae* was detected in all tested samples, and the average relative abundance of *Mycoplasma hyopneumoniae* accounted for 46% of the total abundance of all bacterial species detected (Supplementary Fig. 5c).

We speculated as to whether *Mycoplasma hyopneumoniae* infected the host and survived from antibiotic selection pressure in the respiratory tract through VFGs. We searched for VFGs present in seven *Mycoplasma hyopneumoniae* MAGs. A total of nine VFGs were detected from seven *Mycoplasma hyopneumoniae* MAGs. Six of these nine VFGs, namely, *P159*, *EF-Tu*, the *P97/P102* paralog family, *PDH-B*, *LppT*, and

*P146*, had a functional capacity of cell adhesion. Except for MAG_366 that had a lower genome completeness (50%), the other six MAGs had at least five gene copies belonging to the *P97/P102* paralog family (Fig. 7a).

We further explored whether the nine VFGs in *Mycoplasma hyopneumoniae* MAGs could be horizontally transferred by MGEs. A total of five MGEs, namely, *pEC4115*, *IS91*, *ISBf10*, *tnpA*, and prophage, were detected in seven MAGs. Among these, at least one *tnpA* gene existed in each MAG (Fig. 7b). Several VFGs were found to be located within 10 kb upstream or downstream of MGEs in the same contigs. For example, the *P159* and *P97/P102* paralog family co-occurred with the *IBSf10* in the MAG_6. However, in MAG_115, the P97/P102 paralog family co-occurred with the MGEs *tnpA* and *IS91* (Fig. 7c). More importantly, *tnpA* and *IS91* were significantly related with 20 and 10 ARGs, respectively (Fig. 2). This suggested that besides VFGs, *Mycoplasma hyopneumoniae* may also have the potential to acquire ARGs from other bacterial species. In addition, two prophages were identified in each of the four *Mycoplasma hyopneumoniae* MAGs for which the completeness of genome achieved 99% (Fig. 7b). More importantly, one *P146* and three *P97/P102* paralog family genes were detected in the genome sequences of two prophages in the MAG_16 (Fig. 7c). These results suggested that VFGs could be horizontally transferred between bacterial strains through different types of MGEs and they promoted the infection of *Mycoplasma hyopneumoniae*.

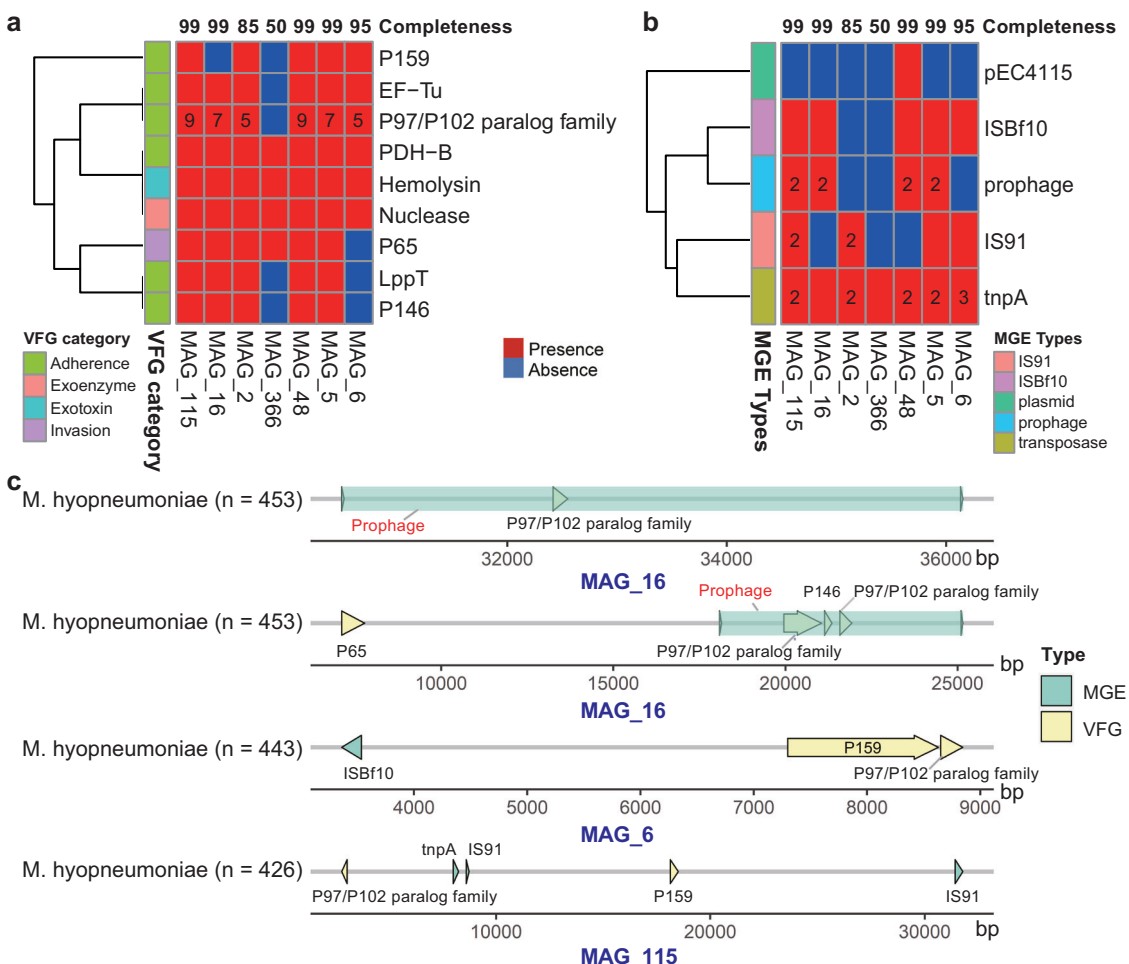

**Fig. 7 | Distribution of virulence factor genes (VFGs) and mobile genetic elements (MGEs) in *Mycoplasma hyopneumoniae* genomes. a** Distribution of VFGs in seven *Mycoplasma hyopneumoniae* MAGs. **b** Distribution of MGEs in seven *Mycoplasma hyopneumoniae* MAGs. Red squares represent the presence of VFGs or MGEs in the genomes, while blue squares represent the missing VFGs or MGEs in the genomes. The numbers of VFGs and MGEs are labeled in the squares if greater than one. The numbers at the top of the heatmap represent the completeness of MAGs. The VFG category and MGE type are indicated with colored boxes in the left.

**c** Distribution of VFGs and MGEs in the contigs of *Mycoplasma hyopneumoniae* MAGs. The *x*-axis represents the locations of genes in the contig. Green arrows indicate MGEs, and yellow arrows indicate VFGs. The directions of arrows represent the strand that on which are located. Right arrows indicate the genes on the forward strand, and left arrows indicate the genes on the reverse strand. The numbers of brackets in the *y*-axis shows the sample numbers detected the corresponding MAG. Source data are provided as a Source Data file.

## Discussion

In this study, we characterized the antibiotic resistome profile of the lower respiratory tract microbiome, investigated the relationships between ARGs and MGEs, and identified host bacteria of ARGs using 745 swine lower respiratory tract metagenomes. To our knowledge, this is the largest catalog of ARGs in the swine lower respiratory tract microbiome, in addition to being larger than corresponding biomes for farm animals and humans. Moreover, this is the first investigation of the relationship between the antibiotic resistome of the swine respiratory microbiome and lung lesions in a genetically varied mosaic population.

The ARG profiles in the swine lower respiratory tract microbiome varied among different individuals. Our results showed that 57% of the ARGs were only present in less than 10% of the samples. Meanwhile, 12 ARGs existed in more than 75% of the samples, including four ARGs to aminoglycosides, three to tetracycline, two to sulfonamide, two to multiple drugs and one to phenicol (Fig. 1c). This is very different from the core resistome of the human respiratory tract microbiome that was dominated by ARGs to beta-lactam, fluoroquinolone, macrolide, and tetracycline in a study based on sputum samples from 85 individuals

with and without chronic respiratory disease[6]. The distinct core ARGs may have been caused by the differences in host sample sources, respiratory sites, utilization of antibiotics, and disease types.

ARGs conferring resistance to tetracycline and aminoglycoside were most abundant in the swine lower respiratory tract microbiome. These ARGs were also frequently detected and had high abundance in fecal and environmental samples of pig farms around the world[15,24]. This may be related to the long-term and widespread use of these two classes of antibiotics in swine production[25]. Compared with the gut antibiotic resistome in pigs, the lower respiratory tract antibiotic resistome had a higher abundance of phenicol resistance genes (Fig. 1b). In particular, the *floR* gene that was carried by various bacterial species accounted for 69% of the total abundance of phenicol resistance genes and had high prevalence (ranked second) in the swine lower respiratory tract (Fig. 1c). This should be related to the use of florfenicol, a fluorinated chloramphenicol derivative that has been commonly used to control respiratory tract infections in pig production. The *floR* gene also had high abundance in swine manure and could be used as an indicator for estimating the total abundance of ARGs[26]. *floR* showed potential human pathogenicity[20] and has been

found in many pathogens, such as *Escherichia coli*[27], *Klebsiella pneumoniae*[28], and *Vibrio cholerae*[29]. This was consistent with the highest abundance of *floR* in pigs with severe lung lesions (Fig. 6d), suggesting that ARGs might be used as a biomarkers for evaluating the lung lesions, and pigs might be used as a model for investigating the relationship between ARGs and lung diseases for humans.

Our data suggested that some MGEs might significantly promote the horizontal transfer of ARGs across different bacterial species. The *tnpA* gene, as a major MGE was widely distributed in various bacterial species, and closely linked to different types of ARGs in all human and pig lung microbiomes, and pig gut microbiome (Fig. 3b). The *Tn916* family was associated with the highest number of ARGs, many of which confer resistance to tetracycline (Fig. 2). Conjugative transposons of the *Tn916* family transfer major antibiotic resistance determinants of many Gram-positive pathogens and are responsible for the dissemination among these pathogens[18]. We did find that tetracycline ARGs related to the *Tn916* family, e.g., *tetM*, *tet(45)*, and *tet(W/N/W)*, were carried by common Gram-positive pathogens including *Enterococcus faecalis, Enterococcus faecium* and *Staphylococcus aureu* (Supplementary Fig. 4 and Supplementary Table 2). Furthermore, the extensive identification of the close linkage between *Tn916* family and *tetM* in all pig lung microbiome, pig gut microbiome, and human lung microbiome implied that MGEs belonging to the *Tn916* family might facilitate to the spread of the *tetM* between animals and humans. The *adeF* gene had the most various host bacteria and the highest prevalence in all swine lower respiratory tract microbiome, pig gut microbiome[15] and human lung microbiome. This might be due to the transfer of *adeF* by various types of MGEs. We did find that *adeF* was significantly associated with 13 MGEs (Fig. 2).

Our results suggested that the α-diversity of the lung antibiotic resistome decreased with the severity of lung disease. This should be explained by the decreased microbial diversity caused by the increased abundance of *Mycoplasma hyopneumoniae* without harboring ARGs. Gammaproteobacteria containing many pathogens, such as *Escherichia coli*, *Pseudomonas aeruginosa*, and *Acinetobacter baumannii* were the dominant ARB in the lower respiratory tract microbiome (Fig. 4a) as well as in the gut[30,31] and the environmental microbiomes[32]. This should be due to 1) the transmission of these ARB across different environments; 2) the transfer of ARGs to these bacteria by MGEs in different environments. Therefore, inhibiting the abundances of these ARB would be the key step to control the spread of ARGs. Antibiotics that could cause antibiotic resistance in these ARB should be used cautiously.

In another our study mentioned above, we identified that *Mycoplasma hyopneumoniae* strains and the adhesion-related virulence factors carried by these *Mycoplasma hyopneumoniae* strains were significantly associated with pig lung lesions[8]. *Mycoplasma hyopneumoniae* must attach to the cilia of respiratory epithelium to infect the host[33]. The P97 adhesin has been shown to play an important role in the pathogenicity of *Mycoplasma hyopneumoniae* and has been treated as a potential vaccine candidate[34,35]. However, another study only observed weak responses against the adhesin P97 C-terminal fragment in analyzing swine antigen-specific antibody responses to *Mycoplasma hyopneumoniae* infection, whereas the responses against the members of the *P97/P102* gene family were strong[36]. The *P97/P102* paralog family members are multifunctional cilium adhesins that promote the pathogenicity of *Mycoplasma hyopneumoniae* by utilizing host surface glycoconjugates and extracellular matrix components[37]. In our results, *P97* was not detected in seven *Mycoplasma hyopneumoniae* MAGs, but multiple *P97/P102* paralog family genes were detected in six out of seven MAGs (Fig. 7a). These results indicated the potential roles of the *P97/P102* paralog family carried by *Mycoplasma hyopneumoniae* in lung lesions. Furthermore, the VFGs associated with adherence, including the *P97/P102* paralog family, *P146*, and *P159*, co-occurred

with different types of MGEs, and the *P97/P102* paralog family and *P146* could be transferred by prophages (Fig. 7c). These results suggested that MGEs may play an important role in the horizontal transfer of VFGs and thus enhance the pathogenicity of *Mycoplasma hyopneumoniae* in lung lesions. MGEs *tnpA* and *IS91* that co-occurred with the *P97/P102* paralog family were significantly related with 20 and 10 ARGs, respectively (Fig. 2). Unexpectedly, no ARGs have been identified in *Mycoplasma hyopneumoniae* genomes. However, considering the existence of *tnpA* and *IS91*, constant monitoring for ARGs in the *Mycoplasma hyopneumoniae* would be necessary in the future.

In conclusion, we constructed the first comprehensive catalog of ARGs in the swine lower respiratory tract microbiome and investigated the potential horizontal transfer of ARGs through analyzing the distribution of MGEs. We also identified the host bacteria of ARGs and evaluated the relationship between ARGs and lung lesions. The results provide a reference for optimizing the use of antibiotics in swine production and help us to better understand the role of the antibiotic resistome of the swine lower respiratory tract microbiome as it affects the lung health of pigs. However, the main limitation of this study was the short length of assembled contigs due to the low rate of high-quality clean sequence reads that was caused by the contamination of host DNA in the sampling. In further studies combining the advantages of the high accuracy of second-generation sequencing data with the long sequence lengths of third-generation sequencing data, contigs and MAGs with higher quality could be obtained. This would be beneficial for obtaining insights into the relationship between antibiotic resistome and mobilome and for the investigation of the host bacteria of ARGs.

## Methods
### Ethical statement
All procedures involved in experimental pigs were conducted according to the guidelines for the care and use of experimental animals established by the Ministry of Agriculture and Rural Affairs of China. The project was also approved by Animal Care and Use Committee (ACUC) in Jiangxi Agricultural University (No. JXAU2011-006).

### Experimental animals and sample collection
A total of 745 lower respiratory tract microbial samples including 670 BAL fluid samples, 74 tracheal lavage fluid samples, and one esophageal lavage fluid sample from 675 experimental pigs were used in this study (Supplementary Data 1). These experimental pigs were from five populations: $F_7$ pigs of a mosaic population ($n = 618$, 264 ♀ and 354 ♂)[38], Erhualian pigs raised in the Changzhou farm ($n = 9$, 4 ♀ and 5 ♂), Berkshire × Licha line pigs from the Dingnan farm ($n = 28$, 22 ♀ and 6 ♂), wild boars ($n = 9$, 3 ♀ and 6 ♂), and Tibetan pigs from the Linzhi farm ($n = 11$, all female). The detailed information about experimental pigs including breed, age, gender, and health is provided in the Supplementary Data 1. All $F_7$ pigs were housed in a uniformed farm of Jiangxi Agricultural University in Nanchang and provided commercial formula feed containing 16% crude protein and 3100 kcal/kg digestible energy and 0.78% lysine. All Erhualian, Tibetan, and Berkshire × Licha pigs were fed with commercial formula feed satisfying the standard pig nutritional requirements. Water was provided ad libitum from nipple drinkers. All lavage samples were obtained by rinsing bronchoalveoli and trachea with sterile phosphate-buffered saline (PBS) immediately after slaughter. Seventy-four tracheal lavage fluid samples and one esophageal lavage fluid sample were obtained from $F_7$ pigs of the mosaic pig population, among which 69 pigs were also collected BAL fluid samples. All pigs did not receive antibiotic treatment for two months prior to harvest for sample collection. Six samples of PBS solution from the same batch and experienced the same process of sampling but not used for lavage was sampled as blank control.

## Phenotyping of lung lesions

Both the anterior and posterior of lungs from each experimental pig were photographed using a digital camera. Six professionally trained panelists scored the level of lung lesions independently according to the following scoring criteria[39]: (i) A total of 13 sections were scored, including six sections from the anterior lung and seven sections from the posterior lung. The sections of anterior lung included left and right apical lobes, left and right cardiac lobes, and left and right diaphragmatic lobes. The sections of posterior lung included an intermediate lobe and the same six sections for anterior lung. (ii) Each section was assigned a score ranging from 0 to 5, corresponding to the proportion of lesion area: 0%, 0–20%, 20–50%, 50–75%, 75–90%, and >90%. (iii) Due to the different area sizes of the apical lobe, cardiac lobe, diaphragmatic lobe and intermediate lobe, different weights of 20% (5% per section × four sections), 20% (5% per section × four sections), 50% (12.5% per section × four sections) and 10% (one section) were assigned to the apical lobe, cardiac lobe, diaphragmatic lobe, and intermediate lobe, respectively. (iv) The lesion score of each lung was equal to the sum of the score for each section multiplied by the corresponding weight. Then, pairwise correlation analyses were performed on the scoring results from six panelists, and the scoring results from three panelists whose correlation coefficients were greater than 0.9 were used to calculate the mean values of the lesion score for each lung that were treated as the phenotypic values. Re-scoring had to be performed if the scoring results that met the requirements were from fewer than three panelists. Based on the final phenotypic values, pigs were divided into four groups: healthy lung (HL, 0 <score ≤ 0.75), slight lung lesions (SLLs, 0.75 <score ≤ 1.50), moderate lung lesions (MLLs, 1.50 <score ≤ 3.00), and severe lung lesions (SVLLs, score > 3.00).

## DNA extraction, metagenomic sequencing, and data analysis

Microbial DNA was extracted using the QIAamp Fast DNA Stool Mini Kit (Qiagen, Hilden, Germany) according to the manufacturer's instructions. The quality of DNA samples was evaluated with a NanoDrop-1000 and electrophoresis using 0.8% agarose gels. All DNA samples that passed quality control were used for constructing the library. The sequencing libraries were constructed according to the manufacturer's procedures (BGI, China). The libraries were sequenced on a DNBSEQ-T7 platform (BGI, China) adopting a 150-bp paired-end sequencing strategy and generating an average of 70.30 G bases of raw sequencing data. The low-quality reads, adapter sequences, and the reads that matched to the pig reference genome assembly (Sscrofa11.1) were filtered from sequencing data. After the quality control, an average of 1.52 G bases of clean data (10.23 million clean sequence reads), ranging from 0.69 to 23.89 G bases was obtained for each sample due to severe contamination by host genomic DNA sequences (Supplementary Fig. 12a). Clean sequence reads were assembled to contigs using MEGAHIT (v1.2.9)[40] by combining single-sample assembly and co-assembly. A total of 19,685,103 contigs with length ≥ 500 bp were obtained from all 745 samples. The average length of contigs was 1136 bp, and the average N50 value was 1626 bp (Supplementary Fig. 12b). Contigs were then used for ORF prediction by Prodigal (v2.6.3)[41]. All predicted protein sequences of ORFs were clustered using CD-HIT (v4.8.1)[42] at 90% identity. After filtering incomplete genes with length <100 bp, the final non-redundant gene catalog containing 10,337,194 genes was used for downstream analysis.

Gene abundance was calculated by aligning high-quality reads from each sample against the gene catalog using BWA (v2.2.1)[43]. The number of reads mapped to each gene was counted using Feature-Counts (v2.0.1)[44]. The relative abundance of genes was calculated using the methods described previously[45]. The effects of sequencing depth and gene length on the abundance were taken into account. Taxonomic annotation was performed by aligning protein sequences of genes to the Uniprot TrEMBL database (https://www.uniprot.org/help/downloads) by DIAMOND (v2.0.12.150)[46] at the threshold of e-value = 10⁻⁵. Search results were parsed by BASTA (v1.3.2.3)[47]. Taxonomic classification of genes was determined under the criteria e-value ≤ 10⁻⁵, the matched sequence length > 25 bp, identity >80%, and the annotation shared by at least 60% of hits. Six PBS solution samples that were treated as blank controls and six samples of mixed regents that were used for library construction and sequencing as sequencing background control samples were also performed metagenomic sequencing. The procedures of bioinformatics analysis for these 12 control samples were as same as that for experimental samples.

We further grouped the contigs into MAGs using the binning tools MetaBAT2 (v2.15)[48], Maxbin (v2.2.7)[49], and CONCOCT (v1.0.0)[50] for single-sample binning. We also conducted a co-binning analysis based on the contigs generated from co-assembly using the tools described above, except CONCOCT that was replaced by VAMB (v3.0.2)[51]. After metagenomic binning, refining, re-assembling, and dereplication, 397 non-redundant MAGs ( < 99% ANI) with ≥50% completeness and ≤10% contamination were generated. Taxonomic annotation of MAGs was classified by GTDB-Tk (v1.7.0)[52]. The genome annotations of MAGs were performed using Prokka (v1.13)[53] with default parameters. The MAGs annotated to *Mycoplasma* and *Ureaplasma* by GTDB-Tk were annotated with the parameter "--gcode 4" because of the different initiation codon. The relative abundance of MAGs was calculated by CoverM (v 0.6.1) (https://github.com/wwood/CoverM).

## Identification and annotation of ARGs, MGEs, and VFGs, and the calculation of their abundances

ARGs were identified by aligning protein sequences of genes from non-redundant gene catalog or MAGs to the CARD (v 3.1.4) using RGI (v 5.2.1)[54] with default parameters. ARGs conferring resistance to at least two drug classes were grouped into the multidrug class, except the ARGs conferring resistance to macrolide, lincosamide, and streptogramin antibiotics that were grouped into the MLS class. We analyzed the distribution of mobile colistin resistance (*mcr*) gene which was first identified in pigs[55] and has broadly spread in environments[56] and nine other clinically important ARGs in 745 tested samples. The nine clinically important ARGs were chosen based on (1) it has been found in clinically relevant pathogens and identified as high-risk ARGs (Q1, top 25%) to human health[20]; (2) It was listed in the top 20 based on the abundances in pig gut microbiome[15] or lung microbiomes; and (3) The seed sequences were available in the SARG (v 2.2) database[57]. The distributions of clinical ARGs in pig lung microbiome was determined by aligning the gene catalog of pig lower respiratory tract microbiome constructed in the current study to the seed sequences of 10 ARGs using BLASTP (v2.12.0)[58] with options "e-value ≤ 10⁻⁵".

Host bacteria of ARGs were determined by taxonomic assignments of the contigs or MAGs on which ARG ORFs were located[20]. The taxonomic annotation of ARG contigs was performed using Kraken2 (v2.1.2)[59] with the default parameters. To further confirm whether *Mycoplasma hyopneumoniae* harbored ARGs, we integrated 23 genomic sequences of *Mycoplasma hyopneumoniae* downloaded from the NCBI RefSeq database (Supplementary Table 3) and seven *Mycoplasma hyopneumoniae* MAGs constructed in this study.

MGEs were identified by aligning protein sequences of genes against the MGE Database "MobileGeneticElementDatabase" created by Parnanen et al. (2018)[16] using DIAMOND (v0.8.36.98)[60] using the criteria of e-value ≤ 10⁻⁵, >80% sequence identity, and >80% query coverage. Under these criteria, we did not identify any MGEs in seven *Mycoplasma hyopneumoniae* MAGs. However, when only the threshold of e-value ≤ 10⁻⁵ was used, we identified four MGEs in these seven MAGs. Considering that the e-value ≤ 10⁻⁵ has been commonly used in the alignments of many other studies by DIAMOND (v0.8.36.98)[61–63], and MGE annotations of these MAGs were highly consistent with each

other, we retained these MGEs for further analysis. The annotation of prophage sequences within *Mycoplasma hyopneumoniae* MAGs was performed using PHASTER's web interface[64]. VFGs were identified through alignments with the Virulence Factor Database (VFDB)[65] using BLASTP (v2.12.0)[58] with options "e-value $\leq 10^{-5}$ and identity $\geq 80\%$." Since multiple genes in the gene catalog might be annotated to the same ARG, MGE, or VFG, the abundances of ARGs, MGEs, and VFGs were calculated by summing the abundances of all members in each category.

### Confirmation of the abundance changes of ARGs, MGEs and MAGs by qPCR

To validate the abundance changes of ARGs, MGEs and MAGs, and the co-abundance relationships between them, we designed the primers for each of three ARGs, two MGEs, and three MAGs harboring these ARGs and MGEs for qPCR using the Primer3 (v. 0.4.0) (Supplementary Table 4). A total of 23 samples containing both the samples detected and undetected the abundances of these ARGs, MGEs and MAGs in the metgenomic sequencing analysis, were selected for qPCR. The 16 S rRNA gene was used as an internal control in the qPCR analysis. qPCR was carried out in triplicate with Power SYBR Green Mastermix (Takara, Japan) on an Applied Biosystems 7900 system using the following program: 95 °C for 5 min; 40 cycles of 95°C for 15 sec, 62°C for 50 sec, and 95 °C for 15 sec; 60 °C for 15 sec. The correlations between the abundance values from qPCR (ΔCt) and the abundances from metagenomic sequencing (FPKM for ARGs and MGEs, and percentage for MAGs), and between ARGs or MGEs, and the MAGs carrying these genes were analyzed with Spearman correlation analysis in R (v4.1.1).

### Identification and annotation of ARGs and MGEs in the pig gut and human lung microbiomes

To investigate the HGTs of ARGs among pig lung microbiome, pig gut microbiome, and human lung microbiome, we used 6339 MAGs recovered from 500 pig gut metagenomes of $F_6$ pigs from the same mosaic population in our previous study (China National GeneBank DataBase with accession code: CNP0000824)[21], 46 metagenomic sequencing data of BAL fluid samples from children with pneumonia[66], and 118 metagenomic sequencing data of BAL fluid samples from adult COVID-19 patients[67]. The information about pig gut and human lung samples, such as age, gender, and health were provided in Supplementary Data 9. Based on the 6,339 MAGs from 500 pig gut metagenomes, gene annotations were performed using Prokka (v1.13)[53] with default parameters. Forty-six and 118 metagenomic sequencing data of BAL fluid samples were downloaded from GenBank repository (accession number SRP119571) and NCBI Sequence Read Archive under project numbers PRJNA687506, respectively. Quality control of metagenomic sequences was performed as described above (the metagenomic analysis of the pig lower respiratory tract microbiome), except the removal of host DNA sequences which used the human reference genome (GRCh38). Then, co-assemblies were also conducted using the same methods as described above. Using the contigs from the co-assembly, gene prediction was conducted with Prodigal (v2.6.3)[41]. ARGs and MGEs were identified by aligning the protein sequences of genes to CARD (v 3.1.4) and "MobileGeneticElementDatabase" databases using the same threshold values using RGI (v 5.2.1)[54] and DIAMOND (v0.8.36.98)[60], respectively, as described in the section of pig lower respiratory tract microbiome. Taxonomic annotation of MAGs was classified by GTDB-Tk (v1.7.0)[52], and taxonomic annotation of contigs were using KRAKEN2 (v2.1.2)[59].

### Confirming the close linkage relationships between ARGs and MGEs

To further verify the co-occurrence relationship between *Tn916* family and tetracycline ARGs, and between the *tnpA* and various types of

ARGs, we downloaded protein sequences and GTF gene annotation files of 3878 genomes of common ARB isolated from humans and pigs from Refseq Database, including 1172 *Escherichia coli*, 529 *Acinetobacter baumannii*, 642 *Pseudomonas aeruginosa*, 976 *Staphylococcus aureus*, 168 *Enterococcus faecalis*, 297 *Enterococcus faecium*, and 94 *Streptococcus suis* (Supplementary Data 5). ARGs and MGEs in these genomes were then annotated using the same method for the identification of ARGs and MGEs in the gene catalog of pig lower respiratory tract microbiome.

### Statistics & Reproducibility

Most of the statistical analyses and visualizations were performed with R (v4.1.1). The distribution of the host bacteria of ARGs at different taxonomic levels was plotted with Sankey diagrams using the *networkD3* R package[68]. The α-diversity of the compositions of ARGs, MGEs, VFGs, and bacterial species including richness (the number of features) and the Shannon index was calculated using the *vegan* R package[69]. The *vegan* R package was also used to perform Procrustes correlation analysis between bacterial species and ARGs. The correlations in the α-diversity between ARGs and MGEs, between ARGs and bacterial species, and between ARGs and MGR were assessed using *Spearman*'s rank correlation and were visualized using the *ggscatter* function in the *ggpubr* R package[70]. Pairwise comparisons of the α-diversity and the composition of the resistome among four pig groups with different lung lesions, and between lung and trachea microbiomes in the $F_7$ population were performed using two-sided Wilcoxon tests and visualized with boxplots or heatmaps. All boxplots were plotted using the *ggpubr* R package[70]. All heatmaps were plotted using the *pheatmap* R package[71]. Chord diagrams of ARG types in four pig groups were generated by the *circlize* R package[72]. Network visualization between ARGs and MGEs was done using Gephi (v0.9)[73] software. The locations and the corresponding directions of ARGs, MGEs, and VFGs on contigs were visualized using *ggplot2*[74] and *gggenes*[75] R packages. No statistical method was used to predetermine sample size. No data were excluded from the analyses. The experiments were not randomized. The investigators were not blinded to allocation during experiments and outcome assessment.

### Reporting summary

Further information on research design is available in the Nature Portfolio Reporting Summary linked to this article.

## Data availability

The metagenomic sequencing data generated in this study have been deposited in the Genome Sequence Archive (GSA) repository under accession code: CRA007668. MAGs and microbial gene catalog used in this study are available in the GSA database under accession codes: GWHBPMO00000000 - GWHBQBU00000000 and OMIX002571 (https://ngdc.cncb.ac.cn/bioproject/browse/PRJCA010893). Source data are provided with this paper.

## Code availability

All codes produced by this project have been deposited in the GitHub repository (https://github.com/zhouyunyan/LungARGs)[76].

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

## Acknowledgements

We thank all members in the National Key Laboratory of Swine Genetic Improvement and Germplasm Innovation for their technical supports in the sample collections. This work was supported by China Agriculture Research System (No. CARS-35) (L. H).

## Author contributions

L.H. and C.C. conceived the study, interpreted the results, and revised the manuscript; Y.Z. performed metagenomic analysis and statistical analysis, visualized the data and wrote the manuscript; J.L. collected the samples, phenotyped lung lesions, and performed metagenomic analysis. F.H. collected the samples, phenotyped lung lesions, and performed qPCR experiment. H.A. designed the experiments and collected the samples. J.G. coordinated the project. The authors read and approved the final manuscript.

## Competing interests

The authors declare no competing interests.
