## [Peer Review File · Nature Communications]

REVIEWER COMMENTS

Reviewer #1 (Remarks to the Author):

The article provides the first overview of ARGs and mobile genetic elements (MGEs) profiles in the swine lower respiratory tract microbiome by using 745 metagenomes from 675 experimental pigs. They also provide the relationships of ARGs with MGEs. Twelve ARGs conferring resistance to tetracycline were related to a MGE Tn916 family, and multiple types of ARGs were related to a transposase gene *tnpA*. However, the analysis, confirmation Experiment and figure presentation may be enhanced.

Major:

1. Some information of experimental pigs should provide, such as age, gender, health and medication etc.
2. Was the average filtered sequence data of each sample 1.52 Gb of clean Reads or 1.52 Gb bases? What was the distribution of filtered sequence data? What was the minimum filtered sequence data?
3. Which the pig reference genome was used to removal of host DNA sequence? Please note in method.
4. The metagenome analysis general needed about 6 G bases data. How to ensure the accuracy of analysis results with such low data? It would be better to perform experimental verification of the major conclusion.
5. The experimental pigs contained different line pigs. And lavage fluid samples contained tracheal, esophageal and BAL. The article just shows the total profile of ARGs, but it would be better if they reveal the profiles of ARGs in different groups and the relationship of the ARG profiles in different groups. It needed some analysis of different groups, such as distribution of ARG number, distribution of ARG abundance, α -diversity, β -diversity, PCoA/NMDS etc.
6. What was distribution of clinically important ARGs in these samples, such like *mcr-1*, *tet(X4)* etc.? Is any ARG variants in these samples?
7. The profiles of MGEs in different groups should be showed, as the suggestion of ARG profiles.
8. The average length of contigs in the current study was only 1,136 bp. How about the average N50 in these samples? What was the distribution of N50?
9. It would be better to reveal the relationships of ARGs with MGEs by using Single Molecule Real-Time (SMRT) Sequencing. The co-abundance relationship analysis and was not enough. They

provided that 12 ARGs conferring resistance to tetracycline were related to a MGE Tn916 family. It may be an especial case that only tetM was verified in two metagenome assembled genomes.

10. It would be better that they isolated some important bacteria and sequenced DNA to verify the MAGs' result, such like Streptococcus.

11. Some information of pig gut samples and human lung samples should provide, such as age, gender, health and medication etc.

12. Was the gut samples and lower respiratory tract samples collect from the same group pig?

13. In Fig3, the length and the coverage were not need to show in the ContigID, but the samples' name/ID or the number of samples contained the contig need to show us. It let us know how many samples contained this characteristic contigs.

14. It would be better to show the profiles of microbiome in different groups before the analysis of host bacteria of ARGs.

15. The article researched horizontal transfer of ARGs, but it ignored the analysis of plasmid.

16. Why try to use antibiotics resistance genes as molecular markers of lung lesions?

17. Why not use microbiome/VFGs as markers of lung lesions? Was any different of microbiome/VFGs in different lung lesion levels?

18. It would be better to verify the abundance changes of ARGs and microbiome by qPCR, such as *Mycoplasma hyopneumoniae*.

19. It would be better to isolated *Mycoplasma hyopneumoniae* and sequenced DNA by SMRT Sequencing.

20. The introduction section introduces pigs as an ideal model for studying human respiratory diseases, but the results do not mention the significance of the present model for studying human respiratory diseases, and the results and discussion section do not answer the question "Can ARG be used as a biomarker for lung diseases?"

21. The relationship with ARGs and lung lesions was an important part of the article. However, the title and abstract did not reflect this part.

Minor:

1. The figures need to be beautified and the style should be consistent. The type size of figure should be consistent.

2. The correlation coefficient should write italic R. And P-value should write italic P. And it should note either Spearman Correlation Coefficient or Pearson correlation coefficient.

3. The "10" should be subscript in Log₁₀.

4. It would be better that n th power operations were uniform. Either n Δ 10m, or n em.
5. The description of sequencing data throughput was inaccurate. Either some million Reads, or some G bases.
6. In Fig4b, it didn't need to prefix before the name, deleted "p_", "c_" etc.
7. Please provide the P-value in Line 341 and 344.
8. It would be better that uniform the " β -lactam" and "beta-lactam".
9. Line 116, "harbor" should be written to "harbors".

Reviewer #2 (Remarks to the Author):

In the paper entitled "Antibiotic resistomes and mobile genetic elements transferred antibiotic resistance genes in pig lower respiratory tract microbiome", Authors characterize the resistome of the pig lower respiratory tract. Their results suggest a link between antimicrobial resistance genes (ARGs) and mobile genetic elements (MGEs), which they also observed in samples from pig gut and human lungs.

Among the merits of the study, it is one of the first studies investigating the lower respiratory tract microbiome of pigs by shotgun metagenomics, providing a large catalogue of ARGs found in the porcine lower airway microbiome. However, it presents potential methodological pitfalls as well as inaccuracies and omissions in the presentation and discussion of the results.

1. One of the main conclusions of the manuscript is that "some MGEs significantly promoted the horizontal transfer of ARGs across different bacterial species" (lines 423-424), which is mainly due to correlations between *tnpA* and tetracycline resistance genes. However, results included in this manuscript are not sufficient to draw this conclusion, since Authors explored the co-abundance of ARGs and MGEs rather than co-occurrence of them in contigs. These results are affected by the small size of assembled contigs, which in turn reflect the high host DNA contamination (approximately 98%).

2. Annotation of contigs harbouring ARGs (paragraph "Host bacteria of ARGs") is also a methodological drawback of the study, especially because of the small contigs size. Upon taxonomical assignment of contigs harbouring ARGs, Authors reported that most of ARGs were harboured by ESKAPE pathogens, including *Pseudomonas aeruginosa*, *Acinetobacter baumannii* and *Escherichia coli*. These are not common colonizers of the porcine lower respiratory tract but rather a result of the taxonomical assignment of small contigs, which in on average consisted in a single gene (average contig size 1,136 bp). When performed on MAGs carrying ARGs (n=115), taxonomical assignation showed different results (compare Fig. 4a with Supplementary Fig. 4d).

3. Although a rigorous evaluation of lung lesion was performed in the paper, Authors only focused on the relationship between ARGs and lung lesion, and no data on differences in microbial composition of the four groups (HL, SLL, MLL and SVLL) are provided. Given the impressive amount of sampling and sequencing performed in this study (745 metagenomes from 675 pigs), the manuscript would benefit of a detailed investigation of the microbial composition of the pig lung microbiome. This could be performed using MAGs and calculating their relative abundance via CoverM (<https://github.com/wwood/CoverM>) or similar tools.

4. Inclusion of negative controls for sampling, DNA extraction and sequencing, and positive controls for DNA extraction and sequencing (mock community) have been proposed as standards in microbiome research (Hornung et al., 2019; doi: 10.1093/femsec/fiz045). Authors should report whether these controls have been included in the study. Lack of these controls, paired with the investigation of relatively low biomass microbiomes (i.e., lung microbiome), provides a source of uncertainty in the results, which are potentially indistinguishable from contaminations, especially for respiratory tract samples with low bacterial densities.

Minor comments

1. The introduction is excessively long, and it should be condensed as much as possible. Lines 64-69 (results of Shuai et al., 2022), lines 95-102 (introduction of MGEs) are some examples of sections that could be condensed.

2. Fig. 1a and 1b are unclear to me. Why ARG number are expressed in % and what is the difference with ARG abundance. Please clarify.

3. Lines 262-263. Please specify the five ARGs with the highest abundance between parenthesis.

4. Lines 287-288. Which is the relative abundance of these three MAGs (340, 368 and 389) among your samples? This can be calculated with CoverM.

5. Lines 473-475. This sentence is too speculative and not supported by results included in the manuscript.

6. The term “superbug” (lines 55 and 242) is not specific for resistant bacteria, and I suggest using multidrug-resistant (MDR) bacteria.

Response letter

Dear reviewers,

Please find the revised version of our manuscript entitled “**Antibiotic resistomes and mobile genetic elements transferred antibiotic resistance genes in pig lower respiratory tract microbiome and its relationships with lung lesions**” (NCOMMS-22-32350A-Z). I and other co-authors appreciate the reviewers very much for your valuable comments. We carefully checked the comments and revised the paper point to point. The revisions on phrase and words were shown in the manuscript, and all revisions were highlighted in blue. The point-by-point responses to the concerns are listed as follows.

Reviewer #1:

The article provides the first overview of ARGs and mobile genetic elements (MGEs) profiles in the swine lower respiratory tract microbiome by using 745 metagenomes from 675 experimental pigs. They also provide the relationships of ARGs with MGEs. Twelve ARGs conferring resistance to tetracycline were related to a MGE Tn916 family, and multiple types of ARGs were related to a transposase gene *tnpA*. However, the analysis, confirmation Experiment and figure presentation may be enhanced.

Response: We have added more bacterial genome data to enhance the analysis and results about the relationships of ARGs with MGEs, especially, about the relationships between ARGs and the MGEs Tn916 and *tnpA*. We have also performed qPCR to confirm the abundances of ARGs and bacterial taxa in tested samples, and the figure presentations have also been enhanced. Please see the more detailed responses about these for comment 4, 9, 10, 13, and 18. The point-by-point response for all comments are listed in the follows.

Major comments:

Comment 1. Some information of experimental pigs should provide, such as age, gender, health and medication etc.

Response: Thanks for this suggestions. The detailed information about experimental pigs including age, gender, breed, health, medication, and the degrees of lung lesions has been provided in the Supplementary Table 1. All pigs did not receive antibiotic treatment for two months prior to harvest for sample collection. This information has been added to the revised manuscript. Please see Lines 622-623.

Comment 2. Was the average filtered sequence data of each sample 1.52 Gb of clean Reads or 1.52 Gb bases? What was the distribution of filtered sequence data? What was the minimum filtered sequence data?

Response: The low-quality reads, adaptor sequences, and the reads that matched the host genomic DNA sequence (Sscrofa11.1) were filtered from sequencing data. After the quality control, an average of 1.52 G bases of clean data, ranging from 0.69 to 23.89 G bases (10.23 million clean sequence reads) was obtained for each sample. The distribution of filtered sequence data is shown in Supplementary Fig. 12a. Please also see in the figure below.

Comment 3. Which the pig reference genome was used to removal of host DNA sequence? Please note in method.

Response: The removal of host DNA sequence was performed using pig reference

genome assembly of Sscrofa11.1. We have added this information to the manuscript. Please see Line 672 in the revised manuscript.

Comment 4. The metagenome analysis general needed about 6 G bases data. How to ensure the accuracy of analysis results with such low data? It would be better to perform experimental verification of the major conclusion.

Response: ≥ 6 G bases data was generally used in the metagenome analysis of gut microbiota and other microecosystem with complex microbial composition. However, it has been difficult to achieve this sequencing depth for microbial samples with low biomass and easily contaminated by host DNA. For example, in this study, an average of 70.30 Gb of raw sequencing data was obtained for each sample. Unfortunately, the vast majority of metagenomic sequence reads were derived from the host genomic DNA. Only 0.69 to 23.89 G bases of clean sequence data (4.65 ~ 159.43 million clean sequence reads) were retained and used for further analyses. Even lower sequence data sizes were obtained in many respiratory tract microbiome studies in humans (Dai, et al., 2019, Bacci, et al., 2017, Xiao, et al., 2022), in which only 1-5% of sequence reads belonging to lower respiratory tract microbiota. For instances, only $453,824 \pm 41,349$ (mean \pm standard error) microbial sequences per sample were obtained and used for the analysis in the study of the microbiota of the airways in cystic fibrosis patients (Bacci, et al., 2017). The study of an integrated human respiratory microbial gene catalogue used a mean of 14,267,332 reads per sample (Dai, et al., 2019). An extremely high host-to-microbial DNA ratio for the lower respiratory tract microbiota samples is a challenge for lung microbiome research (Yi et al., 2022).

As for ensuring the accuracy of analysis results with such low data, to verify the major conclusions, we have performed two aspect additional studies. First, the major results about the relationships of ARGs with MGEs were confirmed by the further data analyses with more sequenced bacterial genomes data from isolated bacterial species downloaded from public databases. 1) Based on 3,878 genomes of common antibiotic resistant bacteria (ARB) from Refseq Database, including *Escherichia coli*, *Acinetobacter baumannii*, *Pseudomonas aeruginosa*, *Staphylococcus aureus*,

Enterococcus faecalis, *Enterococcus faecium*, and *Streptococcus suis*, we found that *tetM* widely coexists with the Tn916 family in various bacterial species, including *Enterococcus faecalis* (existed in 105 of 168 genomes, 105/168), *Enterococcus faecium* (179/297), *Streptococcus suis* (14/94), *Staphylococcus aureus* (124/976), and *Escherichia coli* (38/1,172) (Supplementary Table 7, Supplementary Fig. 4a). Notably, not all strains of these bacteria carried *tetM*, but near all genomes having *tetM* also carried the Tn916 family. This result further confirmed the suggestion that the Tn916 family might play an important role in the horizontal transfer of *tetM*. A total of 78,514 MGE ORFs were identified in these 3,878 genomes, 72.5% of which were *tnpA* that belongs to transposase. This was consistent with the result that transposase genes had the highest abundance (79.0%) in swine lower respiratory tract microbiome (Supplementary Figure 1b). In addition, *tnpA* was close proximity to various types of ARGs (Supplementary Fig. 4b), providing the evidences that *tnpA* might contribute to the HGT of multiple ARGs. Please also see the response for comment 10.

species	Genome number	No. of genomes carrying tetM	No. of genomes carrying both tetM & Tn916	From human	From pig
Streptococcus suis	94	14	14	3	11
Staphylococcus aureus	976	124	124	118	6
Enterococcus faecalis	168	107	105	96	9
Enterococcus faecium	297	180	179	174	5
Escherichia coli	1172	56	38	14	24
Acinetobacter baumannii	529	2	0	0	0
Pseudomonas aeruginosa	642	0	0	0	0
Total	3878	483	460	405	55

Supplementary Fig. 4. The representative of close linkage relationships between antibiotic resistance genes (ARGs) and mobile genetic elements (MGEs) based on 3,878 genomes of isolated common antibiotic resistant bacteria (ARB)

(a) The representative of close linkages between Tn916 family and tetracycline ARGs in several

common antibiotic resistant bacteria (ARB) from pigs and humans. (b) The representative of close linkages between *tnpA* and various ARGs in several common antibiotic resistant bacteria (ARB) from pig and human.

Secondly, to verify the abundance changes and the co-abundance relationships of ARGs, MGEs and microbiome (MAGs) in tested samples, we selected three ARGs, two MGEs, three MAGs carrying these ARGs and MGEs for qPCR analysis in 23 microbial samples that were used in this study, and comprised of both the samples detected and undetected the abundances in the metagenomic sequencing analysis. The results found the significant correlations between the abundance values from qPCR (ΔC_t) and the abundances from metagenomic sequencing for near all ARGs and MGEs, except *tet M* which showed the tendency of correlations but not achieve significance level ($R = 0.35$, $P = 0.099$) (Supplementary Figure 3a). Furthermore, as we expected, the close relationships between ARGs and MGEs, and between ARGs, MGEs and MAGs carrying these genes were well confirmed in the qPCR (Supplementary Figure 3b).

Please also see the response for comment 18.

Supplementary Fig. 3. Confirmation of the abundance changes of ARGs, MGEs and MAGs, and the co-abundance relationships between ARGs, MGEs and MAGs, and between ARGs and MGEs based on qPCR.

(a) Confirmation of the abundance changes of ARGs, MGEs and MAGs in tested samples ($n = 23$) by analyzing the correlations between the abundance values from qPCR (ΔCt) and the abundances from metagenomic sequencing. Because the MAG340 were only detected in three of 23 samples in the metagenomic sequencing, the correlation analysis could not be performed although it was validated in the qPCR. However, MAG21 was not validated. (b) The co-abundance relationships between ARGs, MGEs and MAGs carrying these genes, and between ARGs and MGEs. The co-abundance relationships were well validated for all pairs. The x -axis shows the abundance values by qPCR (ΔCt), and the y -axis represent the abundance values from metagenomic sequencing. The Spearman rank correlation analyses was performed by *ggscatter* function in *ggpubr* R package.

References:

Dai et al., An integrated respiratory microbial gene catalogue to better understand the microbial aetiology of *Mycoplasma pneumoniae* pneumonia, *GigaScience*, 8.8 (2019): giz093.
 Bacci et al., A different microbiome gene repertoire in the airways of cystic fibrosis patients with severe lung disease. *International Journal of Molecular Sciences* 18.8 (2017): 1654.

Xiao et al., Insights into the unique lung microbiota profile of pulmonary tuberculosis patients using metagenomic next-generation sequencing. *Microbiology Spectrum* 10.1 (2022): e01901-21.

Yi et al., The human lung microbiome—A hidden link between microbes and human health and diseases. *iMeta* 1.3 (2022): e33.

Comment 5. The experimental pigs contained different line pigs. And lavage fluid samples contained tracheal, esophageal and BAL. The article just shows the total profile of ARGs, but it would be better if they reveal the profiles of ARGs in different groups and the relationship of the ARG profiles in different groups. It needed some analysis of different groups, such as distribution of ARG number, distribution of ARG abundance, α -diversity, β -diversity, PCoA/NMDS etc.

Response: In this study, we first constructed the catalogs of antibiotics resistance genes (ARGs) and mobile genetic elements (MGEs) of pig lower respiratory tract microbiome with 745 microbial samples from 675 pigs. And then, we described the composition (types and abundances) and distribution (prevalence) of ARGs and MGEs, and the relationships between ARGs and MGEs. We agree that the more information could be presented if the profiles of ARGs in different pig groups and the relationship of the ARG profiles among different groups were described, so we have performed the comparison analysis of ARG and MGE profiles among five pig populations and set this result as a separated section “Comparison of antibiotic resistome and MGE profiles between lung and trachea microbiomes and among five pig populations”.

The results showed that the number of ARGs (richness), the abundance of total ARGs, and the α -diversity (Shannon index) of resistome profile in the trachea microbiome was significantly higher than that in the lung microbiome (Fig. 5a-c), and 78% of ARGs ($n = 292$) had a higher prevalence in tracheal lavage fluid samples compared to BAL fluid samples (Supplementary Table 10). Principal co-ordinates analysis (PCoA) also found the significant difference in the compositions of ARGs between lung and trachea microbiomes (ANOSIM, $R = 0.045$, $P = 0.002$) (Supplementary Figure 7a). We further identified 40 ARGs showing significantly different abundances between

lung and trachea microbiome. Most of these differential ARGs had significantly higher abundance and prevalence in tracheal lavage fluid samples, and mainly conferred the resistance to multiple drugs or aminoglycosides (Fig. 5d). Nine OXA ARGs belonging to beta-lactamases showing resistance to multiple drugs and that were transferred by plasmids were found in most of the tracheal lavage fluid samples but were almost absent in BAL fluid samples (Fig. 5d). The similar results were also identified in the MGE profiles. Significantly different MGE compositions were found between lung and trachea microbiomes by PCoA (Supplementary Figure 7b). Trachea microbiome had the higher number of MGEs (richness), total abundance of MGEs, and the α -diversity (Fig. 5e-g).

We further compared ARGs in pig lower respiratory tract microbiome among five pig populations. Overall, the richness, the total abundance, and the α -diversity (Shannon index) of ARGs were not significantly different among five pig populations (Supplementary Figure 8a-c). However, different pig populations showed distinct β -diversity in the ARG composition. Erhualian pigs had the highest β -diversity (Supplementary Figure 9a). PCoA also indicated the distinct compositions of ARGs among five pig populations (Supplementary Figure 9b). Based on the relative abundances, the F7 population had the highest abundance of *floR. tet(L)* was the dominant ARGs in both Berkshire \times Licha line and wild boars. *tet(39)* and *tetQ* were the dominant ARGs in Erhualian and Tibetan pigs, respectively.

Please see Lines 383-412 in the revised manuscript.

Comment 6. What was distribution of clinically important ARGs in these samples, such like *mcr-1*, *tet(X4)* etc.? Is any ARG variants in these samples?

Response: We analyzed the distribution of nine clinically important ARGs including two tetracycline resistance genes (*tet(L)* and *tet(D)*), one sulfonamide resistance genes (*sulI*), two MLS (macrolide, lincosamide, and streptogramin antibiotics) resistance genes (*ErmB* and *ErmT*), and four aminoglycoside resistance genes (*APH(6)-Id*, *APH(3'')-Ib*, *ANT(6)-Ia*, and *aad(6)*) from the non-redundant ARG catalog of pig lower respiratory tract microbiome constructed in the current study in 745 tested

samples. These nine clinically important ARGs were chosen based on (1) they have found in clinically relevant pathogens and identified as high-risk ARGs (Q1, top 25%) to human health (Zhang, et al., 2022); (2) they were listed in the top 20 based on the abundances in pig gut microbiome (Zhou, et al., 2022) or lung microbiomes (current study); and (3) The seed sequences are available in the SARG database (Yin et al., 2018). Among these nine ARGs, *sull* and *ErmB* were highly variable in pig lower respiratory tract microbiome. Six and seven ORFs were identified as *sull* and *ErmB*, respectively. And the prevalence of these ORFs varied in 745 tested samples (Supplementary Table 4). We also analyzed the variants of mobile colistin resistance (*mcr*) gene which was first identified in pigs and has broadly spread in environments, and found 13 subtypes of *mcr* in pig lung microbiomes, such as *mcr-1*, *mcr-1.2*, *mcr-3.4*, and *mcr-4*. The prevalence of these *mcr* subtypes was different in 745 tested samples (Supplementary Table 4). However, *tet(X4)* was not detected in both pig lung and gut microbiomes.

These results was presented at the revised manuscript, please see Lines 156-166.

References:

Zhang et al., Assessment of global health risk of antibiotic resistance genes. Nat Commun. 2022, 13(1):1553.

Zhou et al., Extensive metagenomic analysis of the porcine gut resistome to identify indicators reflecting antimicrobial resistance. Microbiome. 2022, 10(1):39.

Yin et al., ARGs-OAP v2.0 with an expanded SARG database and Hidden Markov Models for enhancement characterization and quantification of antibiotic resistance genes in environmental metagenomes. Bioinformatics. 2018, 34(13):2263-2270.

Comment 7. The profiles of MGEs in different groups should be showed, as the suggestion of ARG profiles.

Response: As the response for comment 5, in this study, we first constructed the catalogs of antibiotics resistance genes (ARGs) and mobile genetic elements (MGEs) of pig lower respiratory tract microbiome with 745 microbial samples from 675 pigs.

And then, we described the composition (types and abundances) and distribution (prevalence) of ARGs and MGEs, and the relationships between ARGs and MGEs.

Based on this comment, we further analyzed the profiles of MGEs including MGE number (richness), MGE abundance, α -diversity (Shannon index) and β -diversity, and performed PCoA among five pig populations, and between lung (BAL) and trachea (tracheal lavage) fluid samples (Fig. 5e-g, Supplementary Fig. 8d-f, and Supplementary Fig. 9c, d). Overall, there were no significant differences in the richness, the total abundance, and the α -diversity of the MGE compositions among five pig populations although distinct compositions of MGEs were observed through PCoA and β -diversity analysis (Supplementary Fig. 8d-f and Supplementary Fig. 9c, d). Based on their abundances, *tnpA* had the highest abundance in all five populations, followed by *IS91*.

These results were also added to the revised manuscript. Please see Lines 413-418.

Comment 8. The average length of contigs in the current study was only 1,136 bp. How about the average N50 in these samples? What was the distribution of N50?

Response: A total of 19,685,103 contigs with length ≥ 500 bp were obtained from all 745 microbial samples. The longest length of these contigs was 849,460 bp. The average length of contigs was 1,136 bp. The average N50 value was 1,626 bp in 745 samples, ranging from 639 bp to 3,742 bp. The distribution of N50 is shown in the follows. We also added this information to the revised manuscript. Please see lines 677-679.

Comment 9. It would be better to reveal the relationships of ARGs with MGEs by using Single Molecule Real-Time (SMRT) Sequencing. The co-abundance relationship analysis was not enough. They provided that 12 ARGs conferring resistance to tetracycline were related to a MGE Tn916 family. It may be an especial case that only *tetM* was verified in two metagenome assembled genomes.

Response: As the reviewer mentioned, long-read sequencing (SMRT sequencing) could benefit to reveal the relationships of ARGs with MGEs. However, the first thing for long-read sequencing was the isolation and culture of targeted bacterial strains. As we all have known that it is difficult to culture some bacterial strains at present stage, and the isolation and culture of targeted bacterial strains are very time consuming. To answer the reviewer's concern, as an alternative, the utilization of genomic sequences of bacterial strains downloaded from the database is an effective method. Here, to verify the relationships of ARGs with MGEs, we have performed the further data analyses with 3,878 bacterial genomes from isolated and sequenced bacterial species downloaded from the Refseq database. These 3,878 bacterial genomes belong to common antibiotic resistance bacteria (ARB), including *Escherichia coli*, *Acinetobacter baumannii*, *Pseudomonas aeruginosa*, *Staphylococcus aureus*, *Enterococcus faecalis*, *Enterococcus faecium*, and *Streptococcus suis*. We found that *tetM* widely coexists with the Tn916 family in various bacterial species, including

Enterococcus faecalis (existed in 105 out of 168 genomes), *Enterococcus faecium* (179/297), *Streptococcus suis* (14/94), *Staphylococcus aureus* (124/976), and *Escherichia coli* (38/1,172) (Supplementary Table 7, Supplementary Fig. 4a). Notably, not all strains of these bacteria carried *tetM*, but near all genomes having *tetM* also carried the Tn916 family. This result further confirmed the suggestion that the Tn916 family might play an important role in the horizontal transfer of *tetM*. We also found another tetracycline ARG *tet(45)* that was closely adjacent to *tetM* in *Enterococcus faecalis* and *Enterococcus faecium*. Furthermore, *tet(W/N/W)* also co-occurred with the Tn916 family in several strains of *Enterococcus faecalis* and *Enterococcus faecium* (Supplementary Fig. 4a). Based on these results, we suggested that the relationships between *tetM* and the Tn916 family should extensively exist, but not an especial case. As for the fact that the co-occurrence relationships between *tetM* and the Tn916 family were only detected in a few contigs in both human and pig lower respiratory tract metagenome data used in this study, this should be due to the short contig lengths that were caused by the low sequencing depth.

Comment 10. It would be better that they isolated some important bacteria and sequenced DNA to verify the MAGs' result, such like *Streptococcus*.

Response: The first things for whole-genome sequencing of bacteria were the isolation and culture of targeted bacterial strains. As we all have known that it is difficult to culture some bacterial strains at present stage, especially from bronchoalveolar lavage (BAL) fluid, and the isolation and culture of targeted bacterial strains are very time consuming. As the responses to comment 4 and 9, to verify the relationships of ARGs conferring resistance to tetracycline with MGEs, we have performed the additional data analyses with 3,878 bacterial genomes from isolated and sequenced bacterial species belonging to common antibiotic resistance bacteria (ARB) and downloaded from the Refseq database. We verified the co-occurrence relationship between *tetM*, *tet(45)*, *tet(W/N/W)* and Tn916 family in *Streptococcus suis*, *Staphylococcus aureus*, *Escherichia coli*, *Enterococcus faecalis*, and *Enterococcus faecium* (Supplementary Fig. 3a). Furthermore, we also found that *tnpA*

was close proximity to various types of ARGs (Supplementary Fig. 4b), supporting our view that *tnpA* may contribute to the HGT of multiple ARGs.

Comment 11. Some information of pig gut samples and human lung samples should provide, such as age, gender, health and medication etc.

Response: Following this comment, the information, such as age, gender, and health about 500 pig gut microbial samples and 46 human lung microbial samples from children with pneumonia has been provided in Supplementary Table 13. However, the metagenomic sequencing data of lung microbial samples from 118 adult COVID-19 patients were downloaded from public database, and there were no related basic and clinical information that could be obtained from that reference for each individual. We have added this information to the revised manuscript. Please see Lines 771-778.

Comment 12. Was the gut samples and lower respiratory tract samples collect from the same group pig?

Response: Lower respiratory tract microbial samples were collected from F₇ pigs of a mosaic population which was constructed by intercrossing four Western breeds (Duroc, Landrace, Large White, and Pietrain) and four Chinese breeds (Bamaxiang, Erhualian, Laiwu, and Tibetan). The gut microbial samples were collected from F₆ pig of the same mosaic population. Both F₆ and F₇ pigs were raised under uniform indoor conditions at the same experimental farm in Nanchang, Jiangxi Province.

Comment 13. In Fig 3, the length and the coverage were not needed to show in the Contig ID, but the samples' name/ID or the number of samples contained the contig need to show us. It let us know how many samples contained this characteristic contigs.

Response: According to this comment, we have used the name of bacterial species that the contigs were annotated to and the number of samples containing the contig to alternate the length and the coverage in the Contig ID.

More explanations for these contigs are as follows:

The contigs NODE_4_length_6650 and NODE_54_length_8785 carrying the Tn916 family and *tetM* belong to the MAG_21 and MAG_26, respectively (Fig. 3a). These two MAGs were detected in 33 and 161 samples. *TnpA* was close proximity to various types of ARGs. Among them, the contig NODE_97_length_2955 with close proximity relationship between the *tnpA* and *sul2* were from the MAG_75 (Fig. 3b), and this MAG were detected in 123 samples. Another MAG (MAG_340) containing the contig k141_98828_length_26402 harbored nine ARGs and 10 *tnpA* and was detected in 40 samples (Figure 4c).

Comment 14. It would be better to show the profiles of microbiome in different groups before the analysis of host bacteria of ARGs.

Response: In another study with the same metagenomic sequencing data, we constructed comprehensive catalogs for microbial genes and metagenome-assembled genomes of the swine lower respiratory tract microbiome. We explored the profiles of porcine lower respiratory tract microbial compositions in five pig populations and analyzed the relationship of microbial species and its functional capacities (such as virulence factors) with lung lesions (Li *et al.*, Comprehensive catalogs for microbial genes and metagenome-assembled genomes of the swine lower respiratory tract microbiome identify the relationship of microbial species with lung lesions, Nature Communications, under review). However, in the current study, we focused on the ARGs and MGEs, and analyzed the composition and distribution of ARGs and ARG-related MGEs in the pig lower respiratory tract microbiome, host bacteria of ARGs, and putative horizontal transfer of VFGs mediated by MGEs. We have believed that this work on ARGs and its horizontal transfer with MEGs need to be specially focused and separately paid attention to as most of other ARG studies did.

According to this comment, we briefly summarized the profiles of porcine lower respiratory tract microbial compositions before the analysis of host bacteria of ARGs as follows:

A total of 81 phyla, 1,018 genera, and 1,611 species were identified in 745 tested samples. Proteobacteria (44%), Tenericutes (31%), Firmicutes (10%), Bacteroidetes

(6%), and Actinobacteria (4%) were the predominant phyla of swine lower respiratory tract microbiome (Supplementary Fig. 5a). *Mycoplasma* (40%) and *Acinetobacter* (17%) were the two most abundant bacterial genera (Supplementary Fig. 5b). At the species level, *Mycoplasma hyopneumoniae* had the highest abundance in the swine lower respiratory tract microbiome, followed by *Acinetobacter johnsonii*, and *Escherichia coli*. Most of the ESKAPE pathogens including *Enterococcus faecium*, *Staphylococcus aureus*, *Klebsiella pneumoniae*, *Acinetobacter baumannii*, *Pseudomonas aeruginosa*, and *Enterobacter* spp., which have been regarded as the most troublesome pathogens in hospitals due to their high frequency of resistance to antibiotics, were also identified in the swine lower respiratory tract microbiome (Supplementary Fig. 5c). In the current study, we also analyzed the relationships between microbial compositions and ARG compositions in the swine lower respiratory tract microbiome (Fig. 6a, b and Supplementary Fig. 10e-f).

These results have been added to the revised manuscript. Please see Lines 308-322, 424-441.

Comment 15. The article researched horizontal transfer of ARGs, but it ignored the analysis of plasmid.

Response: According to this comment, we have added more results about plasmid to the revised manuscript (Lines 180-183 and Lines 222-225) as follows:

Of the 3,016 ORFs annotated as MGEs, a total of 105 ORFs covering 19 MGE types were identified as plasmid genes. However, these plasmid genes only accounted for 1.0% of the total abundance of all MGEs (Supplementary Fig. 1b and Supplementary Table 5). *repUS12*, a MGE belonging to plasmid, was significantly associated with 6 ARGs including *ANT(4')-Ib*, *ANT(6)-Ia*, *APH(2'')-If*, *ErmT*, *tet(45)* and *tet(L)* (Fig. 2). Among them, *tet(L)* was found to coexist with *repUS12* on the same contig (Supplementary Fig. 2).

Comment 16. Why try to use antibiotics resistance genes as molecular markers of lung lesions?

Response: The composition of antibiotic resistance genes in the gut has been reported to be associated with human diseases. For examples, Shuai *et al.* (2022) found that the composition of the gut antibiotic resistome in humans was associated with the progression of diabetes. Another study by Kovtun *et al.* suggested that ARGs could serve as potential predictors of autism spectrum disorder (Kovtun et al., 2020) (Lines 63-68). With increasing occurrences of respiratory diseases, more and more antibiotics have been used to treat respiratory diseases (Lines 53-57). ARGs harbored in bacteria may affect the therapeutic effect of drugs and lead to the increased lung lesions. Therefore, we hypothesized that ARGs might be related to respiratory diseases or could be used as molecular markers for lung lesions. Furthermore, understanding the composition of the antibiotic resistome in the pig lung microbiome could guide the use of antibiotics in treating swine respiratory diseases.

References:

Shuai et al., Human Gut Antibiotic Resistome and Progression of Diabetes. *Adv Sci (Weinh)* 9, e2104965 (2022).

Kovtun et al., Antibiotic Resistance Genes in the Gut Microbiota of Children with Autistic Spectrum Disorder as Possible Predictors of the Disease. *Microb Drug Resist* 26, 1307-1320 (2020).

Comment 17. Why not use microbiome/VFGs as markers of lung lesions? Was any different of microbiome/VFGs in different lung lesion levels?

Response: As the response for comment 14, in another study with the same metagenomic sequencing data, we constructed comprehensive catalogs for microbial genes and metagenome-assembled genomes of the swine lower respiratory tract microbiome, and systematically explored the relationship of microbial species and its functional capacities, such as VFGs with lung lesions (Li *et al.*, Comprehensive catalogs for microbial genes and metagenome-assembled genomes of the swine lower respiratory tract microbiome identify the relationship of microbial species with lung lesions, 2023, *Nature Communications*, under review). We identified several lung

bacterial taxa and VFGs that were significantly associated with the lung lesions. However, in the current study, we focused on ARGs. To further evaluate the relationship of the diversity and composition of ARGs in the lung microbiome with lung lesions, we compared the profiles of lung resistome under different lung lesion levels using 613 BAL fluid samples from F7 pigs of the mosaic population. We also analyzed the distribution of ARGs in the genomes of *Mycoplasma hyopneumoniae*, a marker species that was identified to be significantly associated with lung lesion levels. However, no ARGs were detected in *Mycoplasma hyopneumoniae* genomes including seven MAGs recovered in this study and 23 genomes downloaded from NCBI RefSeq databases. Interestingly, there were many VFGs in *Mycoplasma hyopneumoniae* genomes. Therefore, we further analyzed the relationships of MGEs with the horizontal transfer of VFGs in *Mycoplasma hyopneumoniae*.

Comment 18. It would be better to verify the abundance changes of ARGs and microbiome by qPCR, such as *Mycoplasma hyopneumoniae*.

Response: Following this comment, to confirm the abundance changes of ARGs, MGEs and MAGs in tested samples and the close relationships between ARGs and MGEs, and between ARGs, MGEs and MAGs carrying these genes, we performed the qPCR for ARGs *tetM*, *APH(6)-Id* and *ANT(3'')-IIa*, MGEs *Tn916-orf13* and *Int-Tn916*, and MAGs MAG_21, MAG_26 and MAG_340 in 23 samples that were detected and undetected the abundances of these ARGs, MGEs and MAGs. The results found the significant correlations between the abundance values from qPCR (ΔC_t) and the abundances from metagenomic sequencing for near all ARGs and MGEs, except *tetM* which showed the tendency of correlations but not achieve significance level ($R = 0.35$, $P = 0.099$) (Supplementary Figure 3a). For three MAGs, the abundance changes were well repeated between qPCR and metagenomic sequencing analysis in 23 samples for MAG_26 and MAG_340 although the MAG_21 did not fit well. Furthermore, as we expected, the close relationships between ARGs and MGEs, and between ARGs, MGEs and MAGs carrying these genes were well confirmed in the qPCR (Supplementary Figure 3b).

Comment 19. It would be better to isolated *Mycoplasma hyopneumoniae* and sequenced DNA by SMRT Sequencing.

Response: In this manuscript, we focused on the ARGs and MGEs, and analyzed the composition and distribution of ARGs and ARG-related MGEs in the pig lower respiratory tract microbiome, host bacteria of ARGs, putative horizontal transfer of VFGs mediated by MGEs, and the relationships of ARGs with lung lesions. However, we did not identify any ARGs in the *Mycoplasma hyopneumoniae* genomes including seven MAGs recovered in this study and 23 entirely sequenced *Mycoplasma hyopneumoniae* genomes downloaded from NCBI RefSeq databases.

Mycoplasma hyopneumoniae is very difficult to isolate because of its slow growth and potential overgrowth with other swine mycoplasmas (Maes *et al.*, 2021). Fortunately, in this study, we reconstructed seven *Mycoplasma hyopneumoniae* genomes (MAGs) with binning approach using metagenomic sequencing data. Among these seven genomes, the completeness of three genomes achieved 99.27% and no contamination was found. All these three *Mycoplasma hyopneumoniae* genomes had the 5S, 16S, 23S rRNA genes, and at least 18 tRNAs that complied with the MIMAG standards for the ‘high quality’ MAG set by the Genomic Standards Consortium.

Reference:

Maes *et al.*, Perspectives for improvement of *Mycoplasma hyopneumoniae* vaccines in pigs. *Vet Res* 52 (1), 67 (2021).

Comment 20. The introduction section introduces pigs as an ideal model for studying human respiratory diseases, but the results do not mention the significance of the present model for studying human respiratory diseases, and the results and discussion section do not answer the question "Can ARG be used as a biomarker for lung diseases?"

Response: As for the significance of the present pig model for studying human respiratory diseases (microbiome), we think this study has provided the following

results (data) that might help the pigs being used as a model for the investigation of human lower respiratory tract microbiome:

1. To our knowledge, the current study has provided the largest catalogs of ARGs and MGEs not only for the study of swine lower respiratory tract microbiome, but also for the investigation of human lower respiratory tract microbiome. Particularly, few metagenomic sequencing analyses were performed for human lower respiratory tract microbiome because the collection of bronchoalveolar lavage (BAL) fluid samples is highly difficult from both humans with lung lesions and the healthy humans in the studies of human lung microbiotas. However, microbial samples can be obtained immediately from pig lungs removed from the thoracic cavity after slaughter, effectively avoiding contamination from the mouth and upper respiratory tract. Furthermore, in another study (Li *et al.*, Comprehensive catalogs for microbial genes and metagenome-assembled genomes of the swine lower respiratory tract microbiome identify the relationship of microbial species with lung lesions, 2023, Nature Communications, under review), we compared the compositions and functional capacities of lung microbiome between pigs and humans by using 118 publicly available metagenomic sequencing data of BAL samples from humans. At the phylum level, the composition of the pig lung microbiome was similar to that of the human lung microbiome both having four predominant phyla Proteobacteria, Bacteroidota, Firmicutes, and Actinobacteria. More than 50% of core microbial genera identified in human lung microbiome (24/47) were also found in the pig lung microbiome. Higher percentage of core functional capacities of human lung microbiome (55.0%) were also identified in pig lung microbiome. Based on the results that the bacterial composition of lower respiratory tract microbial community structured the composition of swine lower respiratory tract resistomes that obtained in this study, we suggested that these catalogs should be used as the reference gene catalogs in the study of ARGs and MGEs in the human respiratory tract microbiome.

2. We identified 73 ARGs showing significantly different abundances among pigs with different severities of lung lesions ($FDR < 0.05$). For example, the *floR* which has been reported to be potential human pathogenicity (Zhang, et al., 2022) and has

been found in many pathogens, such as *Escherichia coli*, *Klebsiella pneumoniae*, and *Vibrio cholerae*, had the highest abundance in pigs with severe lung lesions (**Fig. 6b**). This suggested that ARGs might also be used as biomarkers for evaluating the lung lesions in humans.

3. We found that twelve ARGs conferring resistance to tetracycline were related to a MGE Tn916 family, and multiple types of ARGs were related to a transposase gene *tnpA*. Most of these linkage complexes between ARGs and MGEs (the Tn916 family and *tnpA*) were also observed in pig gut microbiomes and human lung microbiomes, suggesting the high risk of these MGEs mediating ARG transfer to both human and pig health.

Following this comment, instead of the statement that pigs are an ideal model for studying human respiratory diseases, we suggest that “Considering the similar microbiota compositions in the lung microbiome between humans and pigs (Nature Communication, under review), the ARG profiles of pig lower respiratory tract microbiome should provide a reference for human lower respiratory tract microbiome.” Please see Lines 73-76.

The statement of “pigs might be used as a model for studying the MGE-mediated horizontal transfers of ARGs in humans from which it was difficult to obtain microbial samples of lower respiratory tract.” has also added to the result section. Please see Lines 302-305.

In the discussion section, we have also added the discussion “*floR* showed potential human pathogenicity and has been found in many pathogens, such as *Escherichia coli*, *Klebsiella pneumoniae*, and *Vibrio cholerae*. This was consistent with the highest abundance of *floR* in pigs with severe lung lesions (Fig. 6d), suggesting that ARGs might be used as a biomarkers for evaluating the lung lesions, and pigs might be used as a model for investigating the relationship between ARGs and lung diseases for humans.” Please see Lines 530-537.

Reference:

Zhang et al., Assessment of global health risk of antibiotic resistance genes. Nat Commun. 2022

Mar 23;13(1):1553.

Comment 21. The relationship with ARGs and lung lesions was an important part of the article. However, the title and abstract did not reflect this part.

Response: According to this comment, we have added the relationship of ARGs with the severity of lung lesions to the abstract section. Please see Lines 40-43 in the revised manuscript and the follows.

“Although the microbial compositions structured the compositions of ARGs, we identified 73 ARGs whose relative abundances were significantly associated with the severity of lung lesions.”

Also, we revised the title of the manuscript to “Antibiotic resistomes and mobile genetic elements transferred antibiotic resistance genes in pig lower respiratory tract microbiome, and its relationships with lung lesions.”

Minor comments:

Comment 1. The figures need to be beautified and the style should be consistent. The type size of figure should be consistent.

Response: Thanks for indicating these. According to this comment, we have adjusted the Figures. The font size and style in each figure have been uniformed. We hope all the figures would meet the requirements.

Comment 2. The correlation coefficient should write italic R. And P-value should write italic P. And it should note either Spearman Correlation Coefficient or Pearson correlation coefficient.

Response: Thank you for your suggestion. Following this comment, all correlation coefficient and P-value have been written as italic *r* and *P*, respectively, Please see Lines 191-192, 235, 391-392, 435, 437, 439-440, 443, and 447. The correlations in the α -diversity between ARGs and MGEs, between ARGs and bacterial species, and between ARGs and microbial gene richness (MGR) were assessed using *Spearman's* rank correlation. This information has been added to the Methods section. Please see

Lines 815-817.

Comment 3. The “10” should be subscript in Log10.

Response: Thanks for indicating this, we have made the corresponding revisions for these Figures. Please see Fig. 1b, Fig. 5d and Supplementary Fig. 11c.

Comment 4. It would be better that nth power operations were uniform. Either n^{10m}, or n^{em}.

Response: According to this comment, we have uniformed the nth power operations. Please see the revised manuscript.

Comment 5. The description of sequencing data throughput was inaccurate. Either some million Reads, or some G bases.

Response: Thanks for indicating this problem. We have made the corresponding revisions in the manuscript. Please see Lines 670-674.

Comment 6. In Fig4b, it didn't need to prefix before the name, deleted “p_”, “c_” etc.

Response: Following this comment, the prefix “p_”, “c_” etc before the tax name in Fig. 4a, Fig. 4b and Supplementary Fig. 5d have been deleted.

Comment 7. Please provide the P-value in Line 341 and 344.

Response: According to this comment, the P-values have been provided to these lines in the revised manuscript. Please see Lines 456, 447.

Comment 8. It would be better that uniform the “β-lactam” and “beta-lactam”.

Response: Following this comment, we have unified the “β-lactam” and “beta-lactam” to “beta-lactam”. Please see Line 457 and 515 in the revised manuscript.

Comment 9. Line 116, “harbor” should be written to “harbors”.

Response: We have made the corresponding revision in the manuscript. Please see

Line 106.

Reviewer #2:

In the paper entitled “Antibiotic resistomes and mobile genetic elements transferred antibiotic resistance genes in pig lower respiratory tract microbiome”, Authors characterize the resistome of the pig lower respiratory tract. Their results suggest a link between antimicrobial resistance genes (ARGs) and mobile genetic elements (MGEs), which they also observed in samples from pig gut and human lungs.

Among the merits of the study, it is one of the first studies investigating the lower respiratory tract microbiome of pigs by shotgun metagenomics, providing a large catalogue of ARGs found in the porcine lower airway microbiome. However, it presents potential methodological pitfalls as well as inaccuracies and omissions in the presentation and discussion of the results.

Response: We thank the reviewer for summarizing the manuscript correctly and his/her concerns on the potential shortcomings. Combing with the reviewer’s comments, we have improved the manuscript by additional bioinformatics analyses through increasing bacterial genome data, performing the validation experiments of qPCR, and adding sequencing data of negative control samples. Particularly,

1) Additional bioinformatics analyses with 3,878 isolated common antibiotic resistant bacterial (ARB) genomes in humans and pigs supported that the close linkage relationships between the Tn916 family and *tetM*, and between *tnpA* and various types of ARGs extensively existed in different bacterial species, but not an especial case. The results also provided the evidence that *tnpA* might contribute to the HGT of multiple ARGs.

2) High throughput metagenomic sequencing and ARG profile analysis of 12 control samples found three ARGs. However, these three ARGs did not exist in 745 experimental samples. The result suggested that the contaminations are unlikely to have an influence on ARGs profiles of experimental samples.

3) As for the different results of taxonomical assignment between Fig. 4a and Supplementary Fig. 4d (now Supplementary Fig. 6d), it should be caused by different numbers of ARGs used for taxonomical assignment (all 1,228 ARG ORFs identified in this study vs. 416 ARG ORFs harbored in 115 MAGs). Please see the detailed explanation for comment 2.

4) As for the matched microbiome analyses, we have provided an integrative analyses and discussion with the microbiome data presented in the related manuscript (NCOMMS-22-32310A-Z.). Please see the response for Comment 3.

The detailed responses for all concerns are listed in the follows, and the corresponding revisions have been made in the manuscript. We hope all these responses and revisions will meet the requirements.

Comment 1. One of the main conclusions of the manuscript is that “some MGEs significantly promoted the horizontal transfer of ARGs across different bacterial species” (lines 423-424), which is mainly due to correlations between *tnpA* and tetracycline resistance genes. However, results included in this manuscript are not sufficient to draw this conclusion, since Authors explored the co-abundance of ARGs and MGEs rather than These results are affected by the small size of assembled contigs, which in turn reflect the high host DNA contamination (approximately 98%).

Response: As the reviewer mentioned, instead of the co-abundance of ARGs and MGEs, the co-occurrences of ARGs and MGEs in the same contigs should be better to support the close linkage between ARGs and MGEs, and potential MGEs-mediated horizontal transfer of ARGs across different bacterial species. To achieve this aim and verify this major conclusion, we used whole-genome sequencing data of 3,878 isolated common antibiotic resistant bacterial (ARB) strains in humans and pigs and performed the additional validation analysis. The results have supported that the close linkage relationships between the Tn916 family and *tetM*, and between *tnpA* and various types of ARGs extensively existed in different bacterial species. The results also provided the evidence that *tnpA* might contribute to the HGT of multiple ARGs. The detailed information about this is described in the follows and in the revised

manuscript:

"We downloaded 3,878 genomes of isolated common antibiotic resistant bacteria (ARB) in humans and pigs from Refseq Database, including 1,172 *Escherichia coli*, 529 *Acinetobacter baumannii*, 642 *Pseudomonas aeruginosa*, 976 *Staphylococcus aureus*, 168 *Enterococcus faecalis*, 297 *Enterococcus faecium*, and 94 *Streptococcus suis* (Supplementary Table 6). The systematic analysis of the close linkage relationships between the Tn916 family and *tetM*, and between *tnpA* and various types of ARGs revealed that *tetM* widely coexists with the Tn916 family in various bacterial species, including *Enterococcus faecalis* (existed in 105 of 168 genomes, 105/168), *Enterococcus faecium* (179/297), *Streptococcus suis* (14/94), *Staphylococcus aureus* (124/976), and *Escherichia coli* (38/1,172) (Supplementary Table 7, Supplementary Fig. 4a). Notably, not all strains of these bacteria carried *tetM*, but near all genomes having *tetM* also carried the Tn916 family. This result further confirmed the suggestion that the Tn916 family might play an important role in the horizontal transfer of *tetM*. A total of 78,514 MGE ORFs were identified in these 3,878 genomes, 72.5% of which were *tnpA* that belongs to transposase. This was consistent with the result that transposase genes had the highest abundance (79.0%) in swine lower respiratory tract microbiome (Supplementary Figure 1b). In addition, *tnpA* was close proximity to various types of ARGs (Supplementary Fig. 4b)."

species	Genome number	No. of genomes carrying tetM	No. of genomes carrying both tetM & Tn916	From human	From pig
Streptococcus suis	94	14	14	3	11
Staphylococcus aureus	976	124	124	118	6
Enterococcus faecalis	168	107	105	96	9
Enterococcus faecium	297	180	179	174	5
Escherichia coli	1172	56	38	14	24
Acinetobacter baumannii	529	2	0	0	0
Pseudomonas aeruginosa	642	0	0	0	0
Total	3878	483	460	405	55

Supplementary Fig. 4. The representative of close linkage relationships between antibiotic resistance genes (ARGs) and mobile genetic elements (MGEs) based on 3,878 genomes of isolated common antibiotic resistant bacteria (ARB)

(a) The representative of close linkages between Tn916 family and tetracycline ARGs in several

common antibiotic resistant bacteria (ARB) from pigs and humans. (b) The representative of close linkages between *tnpA* and various ARGs in several common antibiotic resistant bacteria (ARB) from pig and human.

As for the high host DNA contamination, undoubtedly, an extremely high host-to-microbial DNA ratio for the lower respiratory tract microbiota samples is a main challenge for lung microbiome studies in both pigs and humans (Yi et al., 2022). In this study, we adopted a deep sequencing depth to improve data size of clean sequence reads. An average of 70.30 Gb of raw sequencing data was obtained for each sample. 0.69 to 23.89 G bases of clean sequence data (4.65 ~ 159.43 million clean sequence reads) were retained and used for further analyses. This clean sequence data size was larger than that were obtained in many respiratory tract microbiome studies in humans (Dai, et al., 2019, Bacci, et al., 2017, Xiao, et al., 2022), in which only 1-5% of sequence reads belonging to lower respiratory tract microbiota.

Comment 2. Annotation of contigs harboring ARGs (paragraph “Host bacteria of ARGs”) is also a methodological drawback of the study, especially because of the small contigs size. Upon taxonomical assignment of contigs harbouring ARGs, Authors reported that most of ARGs were harboured by ESKAPE pathogens, including *Pseudomonas aeruginosa*, *Acinetobacter baumannii* and *Escherichia coli*. These are not common colonizers of the porcine lower respiratory tract but rather a result of the taxonomical assignment of small contigs, which in on average consisted in a single gene (average contig size 1,136 bp). When performed on MAGs carrying ARGs (n=115), taxonomical assignment showed different results (compare Fig. 4a with Supplementary Fig. 4d).

Response: (1) About the methodological drawback of the study, especially because of the small contigs size.

The KRAKEN2 (v2.1.2) used for the taxonomical assignment of contigs in this study has been commonly used for taxonomical annotation of contigs harboring ARGs in many other studies (Sukumar et al., 2023; Zhang et al., 2022). As for the small contigs

size, we agree that the longer contigs are the better for the accuracy of annotation. However, in the metagenomic sequencing analysis, clean sequence reads were first assembled into contigs. And then, the assembled contigs were filtered at a length threshold (the minimum length) that has always been set at > 500 bp for respiratory tract microbiome with low biomass (Dai et al., 2019) and >1000 bp for the gut microbiome with mass biomass (Almeida et al., 2019; Parks et al., 2017). That is to say, the contigs with length > 500 bp or > 1000 bp were used for further analyses including taxonomic annotation. As for the average contig length, it is affected by the filtering threshold of the minimum contig length. The average of 1,000 to 1,500 bp is the normal range for the average contig length in the metagenomic sequencing analysis of both low respiratory tract and gastrointestinal tract microbiome although the average contig length in this study was slightly lower than that reported in animal gastrointestinal tract microbiome because of the low biomass (Please see the following Table).

Host	Site	Average contig length (bp)	Journal	Reference
Ruminants	Gastrointestinal tract	1,392 bp	Microbiome	Xie et al., 2020
Pig	Gut	1,428 bp	Nature Microbiology	Xiao et al., 2016
Pig	Low respiratory tract	1,136 bp	Current study	

(2) About different results of taxonomical assignment between contigs harboring ARGs and MAGs carrying ARGs (n=115) (comparing Fig. 4a with Supplementary Fig. 4d)

The different results of taxonomical assignment between Fig. 4a and Supplementary Fig. 4d (now Supplementary Fig. 6d) should be caused by different numbers of ARGs used for taxonomical assignment (all 1,228 ARG ORFs identified in this study vs. 416 ARG ORFs harbored in 115 MAGs). Please see the detailed explanation below.

In the taxonomical assignment of contigs harboring ARGs (host bacteria of ARGs), a

total of 1,228 ORFs were identified as ARGs in this study. These ARGs were distributed on 1,209 contigs. Among them, 715 contigs (59%) could be annotated to the species level (28 species). *Escherichia coli*, *Acinetobacter baumannii* and *Pseudomonas aeruginosa* carried the largest numbers of ARGs (Fig. 4a). The Figure 4a showed the bacterial taxa carrying more than 5 ARGs.

In the taxonomical assignment of MAGs carrying ARGs, only 416 out of 1,228 ORFs identified as ARGs were identified on 277 contigs of 115 MAGs. This means that more than 60% contigs with ARGs could not be binned into MAGs. The low number of MAGs that could be obtained should be due to uneven species abundances, uneven sequence coverage, high strain diversity and low recovery rates for some phyla (Zhou, et al., 2022). This condition also can be observed in the studies of other microbiomes, such as gut microbiome. In this study, we did not obtain any MAGs belonging to *Acinetobacter baumannii* and *Pseudomonas aeruginosa* which were the two bacterial species harboring the large numbers of ARGs described above. However, as we expected, for *Escherichia coli* and *Acinetobacter johnsoni* which carried the large number of ARGs and obtained MAGs (Supplementary Fig. 5c), the result of the taxonomical assignments between contigs harboring ARGs and MAGs carrying ARGs was highly consistent with each other (Supplementary Fig. 6d).

Because only 59% of contigs (715) could be annotated to the species level (220 species) in the taxonomical assignment of contigs harboring ARGs, we further compared the assignment results between contigs harboring ARGs and MAGs carrying ARGs at the genus level. The assignment results were highly consistent with each other. *Acinetobacter*, *Pseudomonas* and *Escherichia* were the three main host bacterial genera harboring ARGs (Fig. 4a). About 54% of ARGs identified in MAGs were carried by *Escherichia* (3 MAGs, 174 ARG ORFs), *Acinetobacter* (20 MAGs, 53 ARG ORFs), and *Pseudomonas* (2 MAGs, 8 ARGs).

(3) About that the ESKAPE pathogens of *Pseudomonas aeruginosa*, *Acinetobacter baumannii* and *Escherichia coli* that harbored most of ARGs are not common colonizers of the porcine lower respiratory tract but rather a result

of the taxonomical assignment of small contigs.

Currently, to our knowledge, most of the studies about swine lung microbiome were performed by 16S rRNA gene sequencing which generally determined the microbial compositions at the genus level. *Acinetobacter baumannii* and *Escherichia coli* were the main antibiotic resistant bacterial species in *Acinetobacter* and *Escherichia*, respectively. These two bacterial genera have been reported as dominant bacteria genera in both pig lung microbiome and human respiratory tract microbiome (Li et al., 2021; Huang et al., 2018; Liu et al., 2020). *Escherichia coli* is one of the most prevalent species reported in swine lung microbiome (Siqueira et al., 2017). It was also detected in the lung microbiome of pulmonary tuberculosis patients and enriched in the untreated pulmonary tuberculosis group (Xiao et al., 2022). Although *Pseudomonas* has rarely been reported in swine lung microbiome because few metagenomic sequencing analyses have been performed for swine lung microbiome. However, it was widely distributed in human respiratory tract microbiome, including lung microbiota, oral microbiota and nasal microbiota (Pragman et al., 2018; Metwally et al. 2020; Pérez-Cobas et al., 2020). Several studies revealed that *Pseudomonas aeruginosa* was the dominant bacterial species in the sputum of patients with cystic fibrosis or chronic obstructive pulmonary diseases (Bacci et al., 2017; Liu et al., 2020).

References:

- Xie et al., An integrated gene catalog and over 10,000 metagenome-assembled genomes from the gastrointestinal microbiome of ruminants. *Microbiome*. 2021, 9(1):137.
- Xiao et al., A reference gene catalogue of the pig gut microbiome. *Nat Microbiol.*, 2016, 1:16161.
- Li et al., Comparative analysis of the pulmonary microbiome in healthy and diseased pigs. *Mol Genet Genomics*, 2021, 296(1):21-31.
- Huang et al., Microbial communities in swine lungs and their association with lung lesions. *Microb Biotechnol.*, 2019, 12(2):289-304.
- Liu et al., Association of sputum microbiome with clinical outcome of initial antibiotic treatment in hospitalized patients with acute exacerbations of COPD. *Pharmacol Res.*, 2020, 160:105095.

Siqueira et al., Microbiome overview in swine lungs. *PLoS One*, 2017, 12(7):e0181503.

Pragman et al., The lung tissue microbiota of mild and moderate chronic obstructive pulmonary disease. *Microbiome*, 2018, 6(1):7.

Metwally et al., Pediatric lung transplantation: Dynamics of the microbiome and bronchiolitis obliterans in cystic fibrosis. *J Heart Lung Transplant.*, 2020, 39(8):824-834.

Pérez-Cobas et al., Persistent Legionnaires' Disease and Associated Antibiotic Treatment Engender a Highly Disturbed Pulmonary Microbiome Enriched in Opportunistic Microorganisms. *mBio*, 2020, 11(3):e00889-20.

Bacci et al., A Different Microbiome Gene Repertoire in the Airways of Cystic Fibrosis Patients with Severe Lung Disease. *Int J Mol Sci.*, 2017, 18(8):1654.

Xiao et al., Insights into the Unique Lung Microbiota Profile of Pulmonary Tuberculosis Patients Using Metagenomic Next-Generation Sequencing. *Microbiol Spectr.*, 2022, 10(1):e0190121.

Sukumar et al., Development of the oral resistome during the first decade of life. *Nat Commun.* 2023 Mar 9;14(1):1291.

Zhang et al., Assessment of global health risk of antibiotic resistance genes. *Nat Commun.* 2022, 13(1):1553.

Dai et al., An integrated respiratory microbial gene catalogue to better understand the microbial aetiology of *Mycoplasma pneumoniae* pneumonia. *Gigascience.* 2019 Aug 1;8(8):giz093.

Almeida et al., A new genomic blueprint of the human gut microbiota. *Nature.* 2019 Apr;568(7753):499-504.

Parks et al., Recovery of nearly 8,000 metagenome-assembled genomes substantially expands the tree of life. *Nat Microbiol.* 2017 Nov;2(11):1533-1542.

Comment 3. Although a rigorous evaluation of lung lesion was performed in the paper, Authors only focused on the relationship between ARGs and lung lesion, and no data on differences in microbial composition of the four groups (HL, SLL, MLL and SVLL) are provided. Given the impressive amount of sampling and sequencing performed in this study (745 metagenomes from 675 pigs), the manuscript would benefit of a detailed investigation of the microbial composition of the pig lung microbiome. This could be performed using MAGs and calculating their relative

abundance via CoverM (<https://github.com/wwood/CoverM>) or similar tools.

Response: In another study with the same metagenomic sequencing data, we constructed comprehensive catalogs for microbial genes and metagenome-assembled genomes of the swine lower respiratory tract microbiome, and systematically explored the relationship of microbial species and its functional capacities, such as VFGs with pig lung lesions (Li *et al.*, Comprehensive catalogs for microbial genes and metagenome-assembled genomes of the swine lower respiratory tract microbiome identify the relationship of microbial species with lung lesions, 2023, Nature Communications, under review). We identified several lung bacterial taxa and VFGs that were significantly associated with the lung lesions.

However, in this study, we have focused on ARGs, and analyzed the ARG profiles and putative MGEs-related horizontal transfer of ARGs in the pig lower respiratory tract microbiome. Furthermore, to further evaluate the relationship of the diversity, composition, and abundance of ARGs in the lung microbiome with lung lesions, we compared the profiles of lung resistome among four pig groups with different lung lesion levels (HL, SLL, MLL and SVLL) using 613 BAL fluid samples from F₇ pigs of the mosaic population. Based on their relative abundances of ARGs, we indeed found that the abundance of ARGs conferring resistance to phenicol was significantly increased in pigs with severe lung lesions, while the abundance of ARGs for aminoglycosides was decreased with the severity of lung lesions (Fig. 6c). We further identified 73 ARGs showing significantly different abundances among pigs with different severities of lung lesions ($FDR < 0.05$).

We have believed that the works on ARGs profiles, putative MEGs-mediated horizontal transfer of ARGs, and the association of ARGs with lung lesions should need to be specially focused and separately paid attention to investigate just as that most of other studies about ARGs in humans or other farm animals did.

In addition, we have added a briefly summarized description about the profiles of porcine lower respiratory tract microbial compositions before the analysis of host bacteria of ARGs and a brief introduction about the relationship between microbial compositions and lung lesions before the section “**Relationship between ARGs in**

the lung microbiome and lung lesions”. Please see Lines 308-322.

Comment 4. Inclusion of negative controls for sampling, DNA extraction and sequencing, and positive controls for DNA extraction and sequencing (mock community) have been proposed as standards in microbiome research (Hornung et al., 2019; doi: 10.1093/femsec/fiz045). Authors should report whether these controls have been included in the study. Lack of these controls, paired with the investigation of relatively low biomass microbiomes (i.e., lung microbiome), provides a source of uncertainty in the results, which are potentially indistinguishable from contaminations, especially for respiratory tract samples with low bacterial densities.

Response: This is an important question raised by the reviewer. To look into the possible contamination introduced during sample collection, DNA extraction and sequencing, we collected six phosphate-buffered saline (PBS) samples that were from the same batch of PBS solution for lavage as blank controls and six samples of mixed reagents that were used for DNA extraction, library construction and sequencing as DNA extraction and sequencing background control samples. High-throughput sequencing was performed for all these 12 samples.

We analyzed the composition of ARGs in twelve control samples and found that:

- 1) Only three ARGs were identified in twelve control samples, including *adeF*, *TEM-181*, and *TEM-237*. These three ARGs were mainly detected from six blank control samples of PBS solution. Among them, only *adeF* was also identified in 745 experimental samples.
- 2) The *adeF* identified in control samples only contained one ORF, and its corresponding host bacteria was *Brevundimonas*. In 745 experimental samples, a total of 57 ORFs were identified as *adeF*. We blasted the *adeF* ORF identified in control samples with 57 *adeF* ORFs in experimental samples. The highest sequence identity was only 72.3%, indicating different *adeF* ORFs between control samples and experimental samples.

These results suggested that the contaminations are unlikely to have an influence on observed microbiome and ARGs profiles.

We have added this result to the revised manuscript. Please see Lines 130-140.

Minor comments:

Comment 1. The introduction is excessively long, and it should be condensed as much as possible. Lines 64-69 (results of Shuai et al., 2022), lines 95-102 (introduction of MGEs) are some examples of sections that could be condensed.

Response: According to this comment, we have condensed the introduction section of the manuscript as much as possible, including reducing the introduction for the results of Shuai et al. (2022) and the introduction of MGEs. More than 120 words have been deleted from the introduction section. Please see the revised manuscript.

Comment 2. Fig. 1a and 1b are unclear to me. Why ARG number are expressed in % and what is the difference with ARG abundance. Please clarify.

Response: Fig. 1a shows the proportions of the number of unique ARGs (subtypes) in each ARG type in the total number of ARGs. Fig. 1b indicates the proportions of the abundance of ARGs in each ARG type in the total abundance of ARGs (Lines 1026-1028). For example, there were 29 unique ARGs (*tet(39)*, *tet(L)*, *tet(Q)*, etc.) for tetracycline resistance, which accounted for 8% of the total number of ARGs (n = 372). The abundance of tetracycline resistance genes accounts for 33% of the total abundance of all ARGs.

Comment 3. Lines 262-263. Please specify the five ARGs with the highest abundance between parenthesis.

Response: According to this comment, we have added this five ARGs in the revised manuscript.

“We further focused on the distribution of host bacteria for the five ARGs with the highest abundances, including *floR*, *tet(39)*, *tet(L)*, *tet(Q)* and *tet(D)*.”

Please also see Line 341 in the revised manuscript.

Comment 4. Lines 287-288. Which is the relative abundance of these three MAGs (340, 368 and 389) among your samples? This can be calculated with CoverM.

Response: Following this comment, we have calculated the relative abundances of these three MAGs (340, 368 and 389) with CoverM. The results are described in the follows and have added to the revised manuscript. Please see Lines 355-359.

“The relative abundances of MAG_340, MAG_368, and MAG_389 were 0.01%~1.37%, 0.01%~2.79%, and 0.002%~2.23% in the samples detected these MAGs, showing a relatively high abundance although their prevalence was low (only detected in 40, eight and eight out of 745 experimental samples).”

Comment 5. Lines 473-475. This sentence is too speculative and not supported by results included in the manuscript.

Response: The results from this study suggested that MGEs might play an important role in the horizontal transfer of VFGs in *Mycoplasma hyopneumoniae*. For example, MGEs *tnpA* and *IS91* co-occurred with the P97/P102 paralog family in *Mycoplasma hyopneumoniae*. We also found that *tnpA* and *IS91* were significantly related with 20 and 10 ARGs, respectively (Fig. 2). Unexpectedly, no ARGs have been identified in *Mycoplasma hyopneumoniae* genomes. Considering the existence of *tnpA* and *IS91*, constant monitoring for ARGs in the *Mycoplasma hyopneumoniae* would be necessary in the future.

Comment 6. The term “superbug” (lines 55 and 242) is not specific for resistant bacteria, and I suggest using multidrug-resistant (MDR) bacteria.

Response: Thank you for your suggestion. We have made the corresponding revision

in the manuscript. Please see Lines 53 and 324.

We hope that all responses and revisions will meet the requirements. Please feel free to contact me if any further information is required.

Best wishes and Yours sincerely

Prof. Dr. Lusheng Huang

Professor& Director, National Key Lab for Swine genetic improvement, Ministry of Science and technology of China,

Member, Chinese Academy of Sciences,

President, Jiangxi Agricultural University,

President, National Committee for domestic animal genetic resources, Ministry of Agriculture and Rural Areas, China,

NanChang, Jiangxi Province, 330045, China; +86 139 7091 9626

REVIEWERS' COMMENTS

Reviewer #1 (Remarks to the Author):

In this revision, the authors have addressed most of my concerns in the previous version regarding the study design and statistical analysis. The revised manuscript has been improved a lot. Only one minor comment is the version of CARD database should be added in method.

Reviewer #2 (Remarks to the Author):

I would like to thank the Authors for thoroughly addressing my comments in their revision. The quality of the manuscript has significantly improved with the incorporation of qPCR data, WGS analyses on published genomes, and the inclusion of analysis on negative controls.

Response letter

Dear reviewers,

Please find the revised version of our manuscript entitled “**Characterization of the pig lower respiratory tract antibiotic resistome**” (NCOMMS-22-32350B). I and other co-authors appreciate the reviewers very much for your valuable comments. We carefully checked the comments and revised the paper point to point. The revisions were shown in the manuscript, and all revisions were highlighted in blue. The point-by-point responses to the concerns are listed as follows.

Reviewer #1 (Remarks to the Author):

In this revision, the authors have addressed most of my concerns in the previous version regarding the study design and statistical analysis. The revised manuscript has been improved a lot. Only one minor comment is the version of CARD database should be added in method.

Response: We are pleased to know that our revisions have addressed most of the reviewer’s concerns and thank the reviewer for indicating this error. Following this comment, the version of CARD (v 3.1.4) has been added to the revised manuscript Please see Lines 721 and 798.

Reviewer #2 (Remarks to the Author):

I would like to thank the Authors for thoroughly addressing my comments in their revision. The quality of the manuscript has significantly improved with the incorporation of qPCR data, WGS analyses on published genomes, and the inclusion of analysis on negative controls.

Response: We are pleased to know that our revisions have thoroughly addressed the

reviewer's comments.

Thanks again for the reviewers' efforts on our manuscript. Please feel free to contact me if any further information is required.

Best wishes and Yours sincerely

Prof. Dr. Lusheng Huang

Professor& Director, National Key Laboratory of Swine Genetic Improvement and Germplasm Innovation, Ministry of Science and technology of China,
Member, Chinese Academy of Sciences,
President, Jiangxi Agricultural University,
President, National Committee for domestic animal genetic resources, Ministry of Agriculture and Rural Areas, China,
NanChang, Jiangxi Province, 330045, China; +86 139 7091 9626